# RNA-binding deficient TDP-43 drives cognitive decline in a mouse model of TDP-43 proteinopathy

Julie C Necarsulmer[1,2], Jeremy M Simon[3,4,5], Baggio A Evangelista[1,2], Youjun Chen[2], Xu Tian[2], Sara Nafees[2], Ariana B Marquez[6], Huijun Jiang[7], Ping Wang[2], Deepa Ajit[2], Viktoriya D Nikolova[4,8], Kathryn M Harper[4,8], J Ashley Ezzell[9], Feng-Chang Lin[7], Adriana S Beltran[5,6,10], Sheryl S Moy[4,8], Todd J Cohen[1,2,3,11]*

[1]Department of Cell Biology and Physiology, University of North Carolina, Chapel Hill, United States; [2]Department of Neurology, University of North Carolina, Chapel Hill, United States; [3]UNC Neuroscience Center, University of North Carolina, Chapel Hill, United States; [4]Carolina Institute for Developmental Disabilities, University of North Carolina, Chapel Hill, United States; [5]Department of Genetics, University of North Carolina, Chapel Hill, United States; [6]Human Pluripotent Stem Cell Core, University of North Carolina, Chapel Hill, United States; [7]Department of Biostatistics, University of North Carolina, Chapel Hill, United States; [8]Department of Psychiatry, The University of North Carolina, Chapel Hill, United States; [9]Department of Cell Biology & Physiology, Histology Research Core Facility, University of North Carolina, Chapel Hill, United States; [10]Department of Pharmacology, University of North Carolina, Chapel Hill, United States; [11]Department of Biochemistry and Biophysics, University of North Carolina, Chapel Hill, United States

*For correspondence: toddcohen@neurology.unc.edu

Competing interest: The authors declare that no competing interests exist.

**Abstract** TDP-43 proteinopathies including frontotemporal lobar degeneration (FTLD) and amyotrophic lateral sclerosis (ALS) are neurodegenerative disorders characterized by aggregation and mislocalization of the nucleic acid-binding protein TDP-43 and subsequent neuronal dysfunction. Here, we developed endogenous models of sporadic TDP-43 proteinopathy based on the principle that disease-associated TDP-43 acetylation at lysine 145 (K145) alters TDP-43 conformation, impairs RNA-binding capacity, and induces downstream mis-regulation of target genes. Expression of acetylation-mimic TDP-43$^{K145Q}$ resulted in stress-induced nuclear TDP-43 foci and loss of TDP-43 function in primary mouse and human-induced pluripotent stem cell (hiPSC)-derived cortical neurons. Mice harboring the TDP-43$^{K145Q}$ mutation recapitulated key hallmarks of FTLD, including progressive TDP-43 phosphorylation and insolubility, TDP-43 mis-localization, transcriptomic and splicing alterations, and cognitive dysfunction. Our study supports a model in which TDP-43 acetylation drives neuronal dysfunction and cognitive decline through aberrant splicing and transcription of critical genes that regulate synaptic plasticity and stress response signaling. The neurodegenerative cascade initiated by TDP-43 acetylation recapitulates many aspects of human FTLD and provides a new paradigm to further interrogate TDP-43 proteinopathies.

## eLife assessment

Necarsulmer et al describe an interesting new mouse model of TDP-43 proteinopathy in which gene editing was used to introduce a K145Q acetylation-mimic mutation previously shown to impair RNA-binding capacity and induce downstream misregulation of target genes. Mice homozygous for this mutation are **convincingly** shown to display cognitive/behavioral impairment, TDP-43

phosphorylation and insolubility, and changes in gene expression and splicing. This novel mouse model replicates some **important** hallmarks of human frontotemporal lobar degeneration and will be an **important** contribution to the field.

## Introduction

TDP-43 proteinopathies are characterized by the dysfunction and aggregation of transactivation response element DNA-binding protein of 43 kDa (TDP-43), with ~95% of all amyotrophic lateral sclerosis (ALS) and ~50–60% of all frontotemporal lobar dementia (FTLD-TDP) cases harboring TDP-43 pathology (*Neumann et al., 2006*; *Neumann et al., 2007*; *Cairns et al., 2007*; *Hogan et al., 2016*). There is significant neuropathologic and clinical overlap between FTLD and ALS with many individuals developing a mixed phenotype, providing strong evidence for a common FTLD/ALS spectrum of disorders (*Burrell et al., 2016*; *Geser et al., 2010*; *Burrell et al., 2011*; *Giordana et al., 2011*; *Geser et al., 2009*). It is also notable that TDP-43 pathology is abundant in other sporadic neurodegenerative diseases including Alzheimer disease (AD) (*Meneses et al., 2021*; *Tomé et al., 2020*), Limbic-Predominant Age-related TDP-43 Encephalopathy (LATE) (*Besser et al., 2020*; *Nelson et al., 2019*), and Parkinson's disease (PD) (*Poulopoulos et al., 2012*; *Nakashima-Yasuda et al., 2007*). The clinical and neuropathological overlap suggests that common pathogenic mechanisms may link TDP-43 to neurodegeneration (*Gao et al., 2018*; *de Boer et al., 2021*). However, modeling sporadic TDP-43 pathogenesis has been challenging since its expression levels are tightly regulated (*Budini and Buratti, 2011*; *Ayala et al., 2011*), which has precluded a clear separation of TDP-43 disease-related dysfunction from general toxicity resulting from TDP-43 over- or under-expression (*Xu et al., 2010*; *Igaz et al., 2011*; *Yang et al., 2014*; *Kraemer et al., 2010*). Current knock-in models using TDP-43 disease-causing mutations (*Fratta et al., 2018*; *White et al., 2018*; *Huang et al., 2020*; *Stribl et al., 2014*; *Ebstein et al., 2019*) provide valuable insights but may be limited in their application to sporadic disease (*Fratta et al., 2018*; *White et al., 2018*; *Huang et al., 2020*; *Stribl et al., 2014*; *Ebstein et al., 2019*).

Under normal physiological conditions, TDP-43 resides in the nucleus to control RNA processing (RNA splicing, transport, and stability) and gene transcription (*Buratti and Baralle, 2010*; *Tollervey et al., 2011*; *Cohen et al., 2011*). Structurally, nuclear retention is primarily mediated by an N-terminal nuclear localization sequence (NLS) through interactions with α1/β-importins (*Doll et al., 2022*; *Pinarbasi et al., 2018*), and association with nucleic acids is mediated by two tandem RNA recognition motifs (RRM1/RRM2) (*Kuo et al., 2014*; *Lukavsky et al., 2013*), however there is interplay between nucleic acid binding and nuclear localization (*Duan et al., 2022*; *Ayala et al., 2008*). The C-terminal glycine-rich domain (also termed the intrinsically disordered or low complexity domain) mediates protein–protein interactions (*Ayala et al., 2008*; *Buratti and Baralle, 2012*; *Budini et al., 2012*) and harbors most, but not all, familial *TARDBP* mutations that are causative for FTLD/ALS (*Pesiridis et al., 2009*; *Sreedharan et al., 2008*). In sporadic and most familial TDP-43 proteinopathies, TDP-43 undergoes nuclear depletion and concomitant nuclear or cytoplasmic accumulation and aggregation (*Kawakami et al., 2019*; *Neumann, 2009*; *Mackenzie and Neumann, 2016*). Both TDP-43 loss-of-function (e.g., aberrant cryptic splicing) and gain-of-function (e.g., aggregate-induced toxicity) mechanisms have been proposed as drivers of TDP-43 pathogenesis (*Cascella et al., 2016*; *Diaper et al., 2013*; *Lee et al., 2011*; *Vanden Broeck et al., 2014*).

How TDP-43 becomes dysfunctional in sporadic disease remains unresolved, however, aberrant TDP-43 post-translational modifications (PTMs), such as phosphorylation, acetylation, and ubiquitination may play a role. PTMs modulate TDP-43's biochemical properties leading to conformational changes, modulation of nucleic acid-binding affinity, regulation of liquid–liquid phase separation, and propensity to form insoluble TDP-43 aggregates, all of which are disease-associated phenomena (*François-Moutal et al., 2019*; *Sternburg et al., 2022*; *Buratti, 2018*). Among the various PTMs, TDP-43 acetylation at lysine residue 145 (K145) within RRM1 has emerged as a critical regulator of both loss- and gain-of-function toxicity (*Cohen et al., 2015*; *Wang et al., 2017*). Acetylated TDP-43 is found within TDP-43 inclusions of sporadic ALS spinal cord motor neurons but not age-matched control tissue (*Cohen et al., 2015*). Inclusions in FTLD cortex are largely composed of C-terminal fragmented TDP-43 lacking the K145 residue (*Igaz et al., 2008*; *Chhangani et al., 2021*), precluding an assessment of Ac-K145 in FTLD patients. However, *TARDBP* mutations that disrupt RNA binding, and

thereby may act in a similar manner to TDP-43 acetylation, have been identified in FTLD-TDP patients (e.g., P112H and K181E) (**Agrawal et al., 2021**; **Chen et al., 2019**), supporting a pathogenic role for altered nucleic acid binding in disease. Mimicking TDP-43 acetylation with a lysine-to-glutamine substitution (TDP-43$^{K145Q}$) is sufficient to neutralize the positive charge, disrupt RNA binding, and induce several hallmarks of TDP-43 pathology in vitro (**Cohen et al., 2015**; **Wang et al., 2017**), supporting a model whereby TDP-43 acetylation drives both loss-of-function (e.g., RNA-binding deficiency) and gain-of-function (e.g., aggregation) toxicity.

Here, we used CRISPR/Cas9 genome editing to introduce a K145Q substitution into the endogenous mouse *Tardbp* gene, thereby generating acetylation-mimic TDP-43$^{K145Q}$ knock-in mice**,** which enabled us to investigate the pathophysiological impacts of an aberrant TDP-43 PTM while leaving native upstream and downstream genomic elements intact. Using mouse cortical neurons, human hiPSC-derived cortical neurons, and aged cohorts of TDP-43$^{K145Q}$ homozygous mice, we found that acetylation-mimic TDP-43$^{K145Q}$-induced nuclear TDP-43 foci and cytoplasmic TDP-43 accumulation, which coincided with several disease-associated and loss-of-function measures including widespread transcriptome and splicing alterations. Finally, we observed prominent FTLD-like cognitive and behavioral deficits in acetylation-mimic TDP-43 mice that correlated with biochemical and splicing alterations in affected brain regions. Our study supports lysine acetylation of TDP-43 as a driver of dysfunction in sporadic TDP-43 proteinopathies.

## Results

### Mouse neurons expressing TDP-43$^{K145Q}$ undergo stress-dependent formation of nuclear TDP-43 foci and loss of TDP-43 splicing function

We originally showed that TDP-43 acetylation can promote RNA-binding deficiency, aggregation, and pathology (**Cohen et al., 2015**; **Wang et al., 2017**). We sought to expand these findings by exploring the behavior of acetylation-mimic TDP-43 variants in primary murine cortical neurons. We employed lentiviral vectors that encode either wild-type (TDP-43$^{wt}$), acetylation-deficient (TDP-43$^{K145R}$), and acetylation-mimic (TDP-43$^{K145Q}$) variants to overexpress TDP-43 species in neurons, and then examined their subcellular localization by immunofluorescence microscopy. In the absence of acute cellular stress, most neurons overexpressing TDP-43$^{K145Q}$ showed distinct nuclear foci that were more prominent compared to TDP-43$^{wt}$ or TDP-43$^{K145R}$ constructs (**Figure 1A, B**). When neurons were exposed to an acute oxidative stressor (200 µM sodium arsenite), a sensitizing agent commonly used to enhance TDP-43 dysfunction (**Dewey et al., 2011**; **Colombrita et al., 2009**; **Gasset-Rosa et al., 2019**; **Cohen et al., 2012**), there was a significant increase in TDP-43 foci formation with all variants (**Figure 1A, C**). The response in neurons expressing TDP-43$^{K145Q}$ was very robust, resulting in the formation of numerous large, bright TDP-43-positive foci, as well as smaller TDP-43-positive nuclear foci. These aberrant TDP-43 structures were absent from cells expressing TDP-43$^{wt}$ or acetylation-null TDP-43$^{K145R}$, indicating that acetylation-mimic TDP-43$^{K145Q}$ alters TDP-43 conformation within the nucleus and sensitizes neurons to oxidative stress-induced foci formation. By coupling high content wide-field microscopy with quantitative image analysis, we observed a threefold increase in TDP-43 foci formation in neurons expressing TDP-43$^{K145Q}$ (**Figure 1C**).

TDP-43 overexpression can result in general toxicity and altered TDP-43 function, depending on the duration and the extent of overexpression (**Xu et al., 2010**; **Ash et al., 2010**; **Yang et al., 2022**). To avoid confounding non-specific toxicity, we transitioned to a more physiologically relevant model to further elucidate the impact of RNA-binding deficient acetylation-mimic TDP-43. We employed CRISPR-Cas9 mutagenesis to introduce a single amino acid substitution at position 145 (K145Q) into the endogenous mouse *Tardbp* locus, thereby generating TDP-43$^{K145Q}$ knock-in mice (**Figure 2— figure supplement 1B**). By targeting the native mouse gene, we avoided both TDP-43 overexpression and disruption of the *Tardbp* untranslated regions (**Budini and Buratti, 2011** ). A TDP-43$^{K145Q}$ founder line was propagated as heterozygotes and continually re-sequenced to confirm retention and propagation of the K145Q substitution (**Figure 2—figure supplement 1C, D**). Both heterozygous and homozygous TDP-43$^{K145Q}$ mice were born at normal mendelian frequencies and showed no obvious developmental defects.

We first investigated the effects of TDP-43$^{K145Q}$ expression in neurons in vitro by isolating and culturing primary cortical neurons from homozygous TDP-43$^{K145Q}$ mice, hereafter referred to as

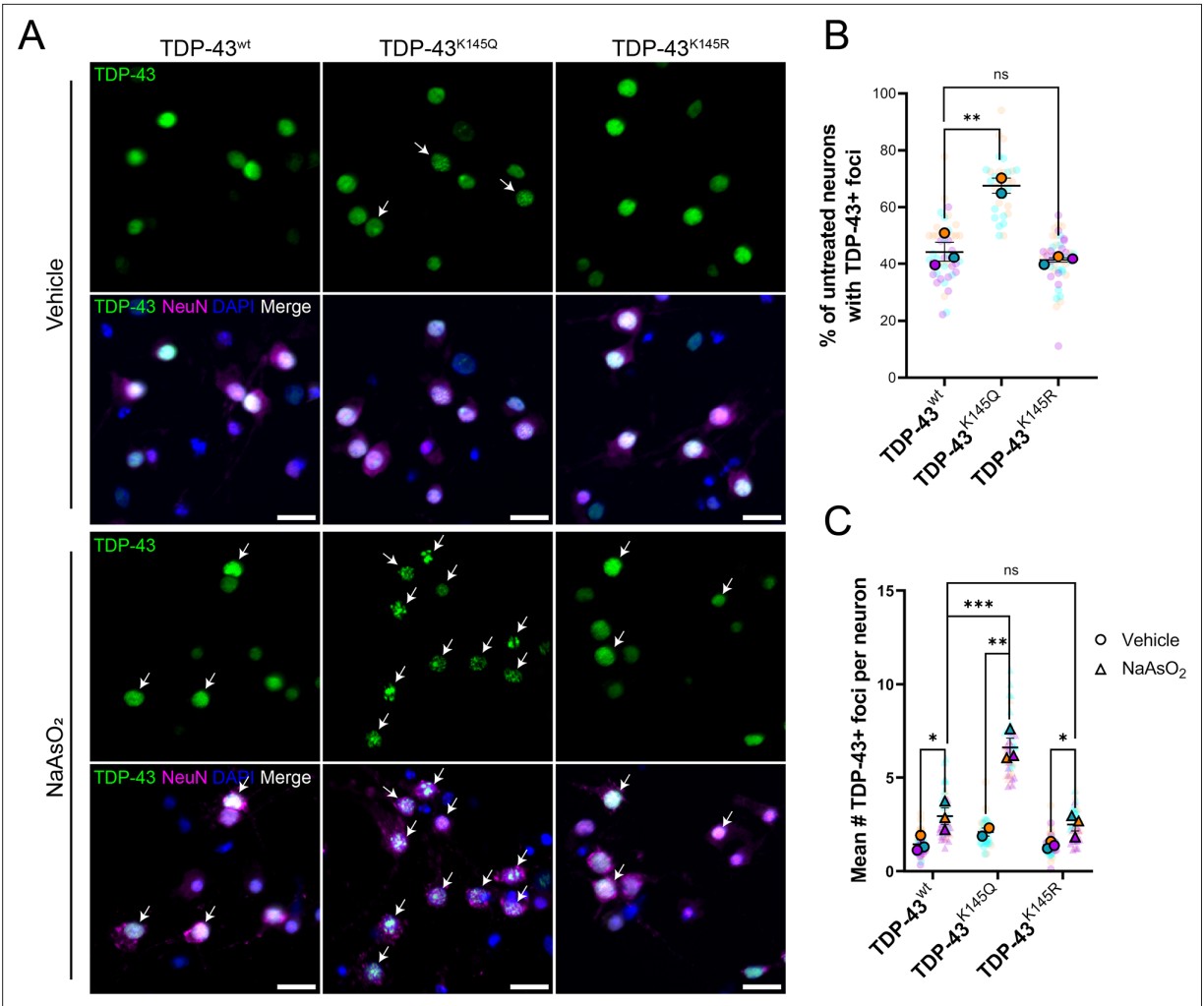

**Figure 1.** Ectopic expression of acetylation-mimic TDP-43[K145Q], an RNA-binding deficient mutant, shows accelerated formation of stress-induced nuclear TDP-43 foci. (**A**) Representative immunofluorescence images of TDP-43 in DIV14 mouse primary cortical neurons overexpressing TDP-43[wt], TDP-43[K145Q], or TDP-43[K145R] after vehicle or 200 µM NaAsO$_2$ treatment followed by labeling of TDP-43 (green), NeuN (magenta), and DAPI (blue). Arrows highlight nuclei with TDP-43+ foci. Scale bar = 20 µm. (**B**) Quantification of percentage of neurons with TDP-43+ foci in vehicle-treated neurons. (**C**) Quantification of the average number of TDP-43+ foci per neuron. Data shown as Superplots (***Lord et al., 2020***); solid color bordered symbols and error bars indicate mean value of each biological replicate ± standard error of the mean (SEM); semi-transparent datapoints represent the average value per neuron in a single field of view, 10–110 neurons per field, 48 fields across $n$ = 2–4 biological replicates. Statistical analysis completed using a linear mixed-effect model. Statistical significance is represented by asterisks *p < 0.05, **p < 0.01, ***p < 0.001. DIV = day in vitro.

TDP-43[KQ/KQ] mice and compared them to TDP-43[wt]-derived neurons. Exposing neurons to acute oxidative stress induced more abundant TDP-43-positive nuclear foci in acetylation-mimic TDP-43[KQ/KQ] neurons than in TDP-43[wt] neurons (***Figure 2A, B***). Quantitative image analysis also revealed variable levels of nuclear clearing and cytoplasmic mislocalization of TDP-43 in TDP-43[KQ/KQ] neurons, however, no statistically significant differences in TDP-43 localization were found between acetylation-mimic and TDP-43[wt] neurons (***Figure 2A, C***). Because TDP-43 foci formation is associated with loss-of-function defects (***Garcia Morato et al., 2022***; ***Mann and Donnelly, 2021***; ***Yu et al., 2021***), we next investigated if TDP-43 function was impaired by evaluating endogenous targets of TDP-43 activity. We observed a trend toward increased *Tardbp* mRNA in TDP-43[KQ/KQ] neurons at DIV14 (***Figure 2D***), which was suggestive of TDP-43 loss of function and auto-regulation. We therefore investigated sortilin-1 (*Sort1*) mRNA splice variants, as a more sensitive endogenous indicator of loss of TDP-43-dependent splicing function. Functional nuclear TDP-43 results in the production of a mature spliced *Sort1* mRNA transcript, however, in the setting of TDP-43 depletion or loss of function, TDP-43 is unable to repress the inclusion of exon 17b, generating a longer *Sort1+ex17b* transcript variant (***Prudencio et al., 2012***;

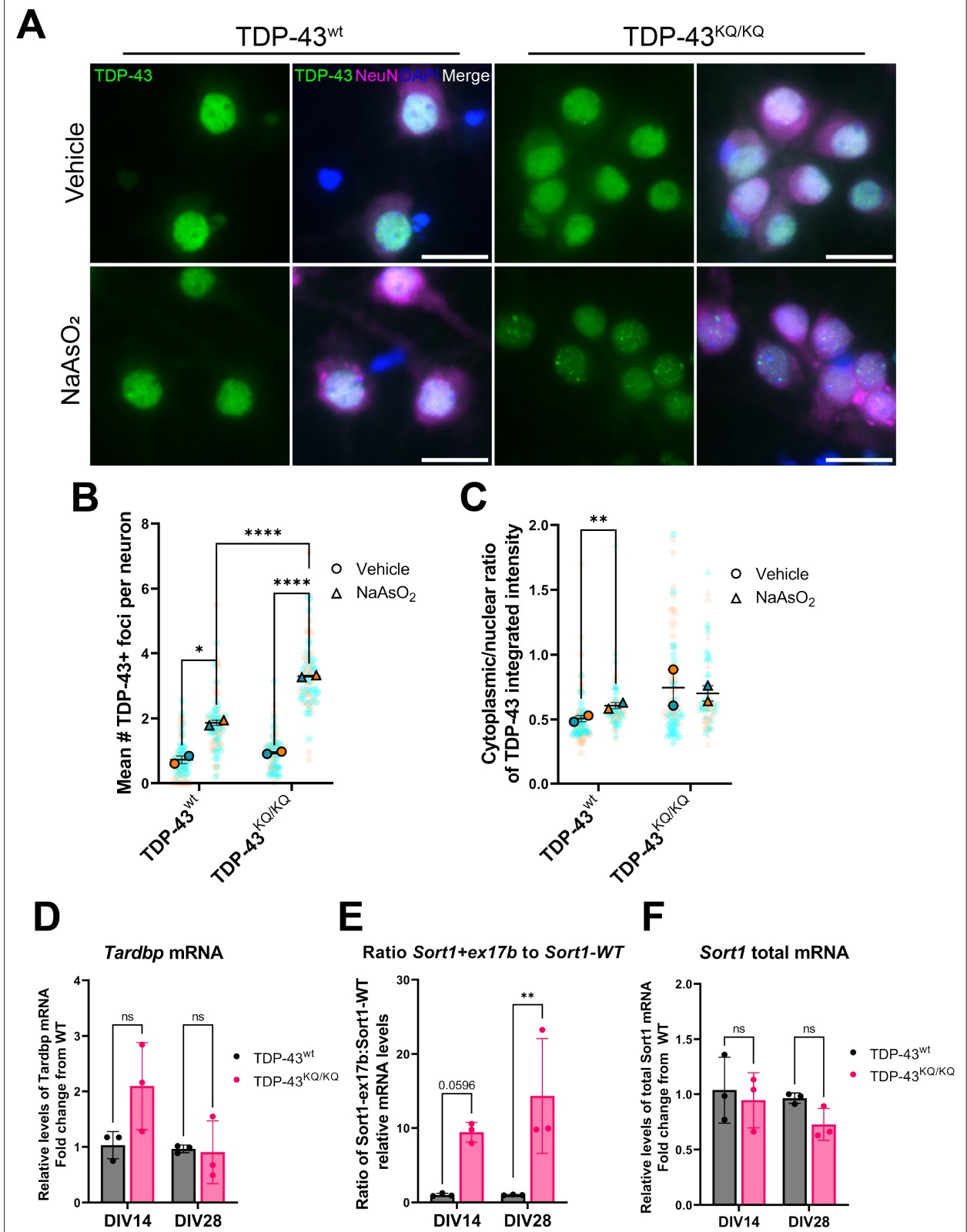

**Figure 2.** An endogenously encoded acetylation-mimic TDP-43$^{K145Q}$ mutation causes altered TDP-43 localization and functional splicing deficits in mouse primary cortical neurons. (**A**) Representative images of primary cortical neurons derived from TDP-43$^{wt}$ or TDP-43$^{KQ/KQ}$ mice that were treated with vehicle or 200 µM NaAsO$_2$ at DIV14 and immunolabeled for endogenous TDP-43 (green), NeuN (magenta), and DAPI (blue). Quantification of the number of TDP-43+ foci (**B**) and the cytoplasmic:nuclear ratio of TDP-43 fluorescence intensity (**C**) in TDP-43$^{KQ/KQ}$ compared to TDP-43$^{wt}$ neurons. RT-

*Figure 2 continued on next page*

Figure 2 continued

qPCR analysis of DIV14 and DIV28 untreated mouse primary cortical neurons using primers specific for mouse (**D**) *Tardbp* [$F_{(1,8)} = 3.034$, $p = 0.1197$]; and *Sort1* splice variants: (**E**) ratio *Sort1+ex17b:Sort1-WT* [$F_{(1,8)} = 23.02$, $p = 0.0014$] and (**F**) total *Sort1* mRNA [$F_{(1,8)} = 5.086$, $p = 0.0541$]. (**B, C**) Data shown as Superplots (*Lord et al., 2020*); solid color bordered symbols and error bars indicate mean value of each biological replicate ± standard error of the mean (SEM); semi-transparent datapoints represent the average value per neuron in a single field of view, 10–110 neurons per field, 72 fields across $n = 2$–4 biological replicates; one color represents one biological replicate. Statistical analysis performed using a linear mixed-effect model. (**D–F**) Analysis by two-way analysis of variance (ANOVA) followed by Šídák's multiple comparisons testing. *F* statistics represent main effect of genotype. Statistical significance is represented by asterisks, *$p < 0.05$, **$p < 0.01$, ****$p < 0.0001$, ns = not significant.

The online version of this article includes the following source data and figure supplement(s) for figure 2:

**Figure supplement 1.** Generation, characterization, and genotyping of TDP-43$^{K145Q}$ knock-in mice.

**Figure supplement 1—source data 1.** Figures showing labeled and unlabeled versions of the uncropped DNA gel image presented in *Figure 2—figure supplement 1*.

*Tann et al., 2019*). RT-qPCR analysis revealed a significant deficit in normal TDP-43 splicing function, with nearly a 10-fold increase in the ratio of *Sort1+ex17b* to *Sort1*-WT mRNA, without significant changes to total *Sort1* mRNA levels in TDP-43$^{KQ/KQ}$ neurons compared to controls (*Figure 2E, F*). Overall, these results indicate that a single endogenously expressed acetylation-mimic TDP-43$^{K145Q}$ mutation can sensitize TDP-43 to conformational changes that impair its normal splicing function in mouse neurons.

## Acetylation-mimic TDP-43$^{K145Q}$ alters TDP-43 function in human neurons

To assess this model's relevance to human neurons, we used CRISPR/Cas9 to generate a panel of hiPSC lines harboring homozygous acetylation-mimic TDP-43 (TDP-43$^{K145Q.12}$ and TDP-43$^{K145Q.18}$), acetylation-deficient TDP-43 (TDP-43$^{K145R.2}$ and TDP-43$^{K145R.12}$), or unmodified TDP-43 (isogenic control, TDP-43$^{wt}$) (*Figure 3—figure supplement 1*). We confirmed appropriate editing of the *TARDBP* gene in each line via Sanger sequencing (*Figure 3—figure supplement 1B*), verified the pluripotency of the hiPSC clones (*Figure 3—figure supplement 1C–E*), and then differentiated each of the lines into mature cortical neurons (*Figure 3—figure supplement 2*). We then used immunofluorescent labeling and confocal microscopy to qualitatively assess the distribution and morphology of TDP-43 protein within neurons, both with and without an acute exposure to sodium arsenite. Untreated hiPSC-derived TDP-43$^{K145Q}$ cortical neurons were morphologically identical to TDP-43$^{wt}$ and TDP-43$^{K145R}$ neurons and showed similar patterns of TDP-43 localization (*Figure 3A*). All hiPSC-derived lines showed a granular nuclear TDP-43 localization pattern under normal conditions, consistent with physiologic de-mixing of nuclear TDP-43 (*Gasset-Rosa et al., 2019*). Following acute oxidative stress, however, TDP-43$^{K145Q}$ neurons showed apparent TDP-43 nuclear clearing and the formation of large, intensely labeled TDP-43-positive foci (*Figure 3A*). In comparison, cortical neurons expressing TDP-43$^{wt}$ or TDP-43$^{K145R}$ maintained nuclear TDP-43 and formed small stippled TDP-43 puncta. We note that TDP-43 nuclear clearing and foci formation was more robust in hiPSC-derived neurons compared to mouse neurons, suggesting human neurons may be more sensitive to TDP-43 RNA-binding deficiency. Similar to mouse neurons, hiPSC-derived TDP-43$^{K145R}$ neurons were indistinguishable from TDP-43$^{wt}$ neurons, supporting acetylation-induced charge neutralization as a driver of TDP-43 loss of function rather than an inherent effect of mutation at the K145 locus.

We next asked whether hiPSC-derived TDP-43$^{K145Q}$ cortical neurons show more robust functional deficits compared to the mouse neuron data above. We again employed RT-qPCR to measure levels transcripts associated with TDP-43 loss of function, beginning with *TARDBP* as an indicator of impaired TDP-43 autoregulation. In contrast to mouse neurons, we observed an approximately twofold increase *TARDBP* mRNA in TDP-43$^{K145Q.12}$ and TDP-43$^{K145Q.18}$ neurons and a slight but significant reduction in *TARDBP* mRNA in TDP-43$^{K145R.12}$ neurons (*Figure 3B*). We then assessed TDP-43-dependent splicing activity using primers specific for human *SORT1* splice variants, whose regulation by TDP-43 is conserved in mouse and humans (*Prudencio et al., 2012*; *Tann et al., 2019*). The aberrant *SORT1+ex17b* splice variant was undetectable in TDP-43$^{wt}$ neurons, but was present in 60% of TDP-43$^{K145Q.12}$ and 100% of TDP-43$^{K145Q.18}$, TDP-43$^{K145R.2}$, and TDP-43$^{K145R.12}$ neurons (*Figure 3C*). In samples confirmed to have detectable *SORT1+ex17b* transcript, levels were, on average, considerably higher levels in TDP-43$^{K145Q}$ compared to TDP-43$^{K145R}$ neurons. Appropriately spliced *SORT1*-WT and total *SORT1* transcript levels were comparable across all five lines, with only a slight increase in

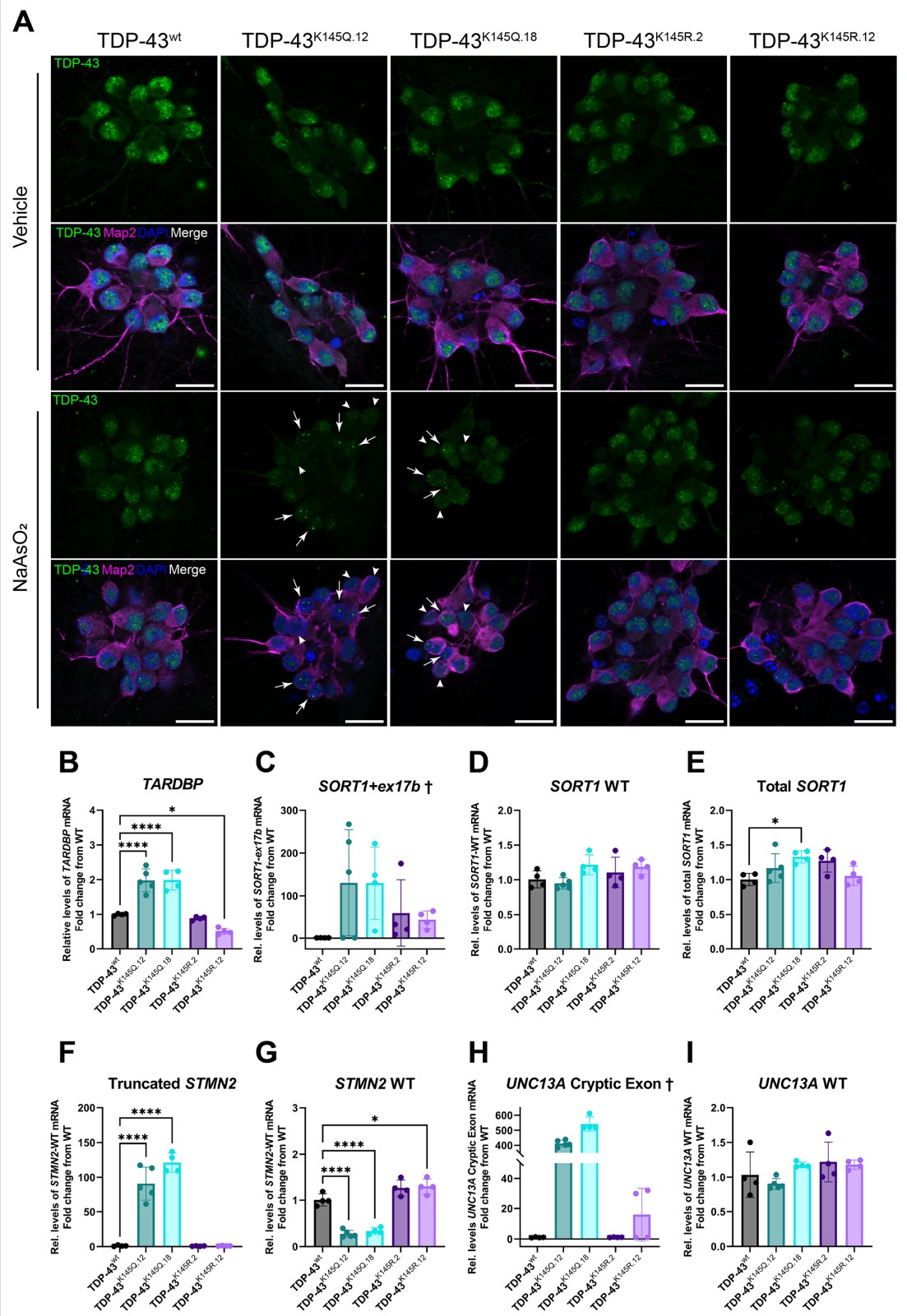

**Figure 3.** RNA-binding deficient TDP-43^K145Q in human-induced pluripotent stem cell (iPSC)-derived mature cortical neurons alters TDP-43 distribution and causes disease-relevant splicing dysregulation. (**A**) Representative confocal images of CRISPR-modified human iPSC-derived cortical neurons harboring homozygous TDP-43^K145Q (clonal lines 12, 18) or TDP-43^K145R (clonal lines 2, 12) knock-in mutations or an isogenic wild-type TDP-43^wt control lacking *TARDBP* modifications. Differentiated cortical neurons were treated with vehicle or 200 µM NaAsO$_2$ and then immunolabeled for endogenous

*Figure 3 continued on next page*

*Figure 3 continued*

TDP-43 (green), Map2 (magenta), and DAPI (blue). Scale bars = 20 μm. Arrows highlight nuclei with few, bright TDP-43+ foci. Arrowheads highlight nuclei with relative depletion of TDP-43 intensity. RT-qPCR analysis of hiPSC-derived mature cortical neuron samples using primers specific for (**B**) human *Tardbp* [$F = 43.79$, $p < 0.0001$], (**C**) *SORT1+ex17b*, (**D**) *SORT1*-WT [$F = 2.975$, $p = 0.0516$], (**E**) total *SORT1* mRNA [$F = 3.461$, $p = 0.0321$], (**F**) truncated *STMN2* [$F = 79.06$, $p < 0.0001$], (**G**) *STMN2* WT [$F = 63.93$, $p < 0.0001$], (**H**) *UNC13A* cryptic exon, and (**I**) *UNC13A* WT mRNA [$F = 2.116$, $p = 0.1262$]. RT-qPCR results compared via ordinary one-way analysis of variance (ANOVA) (*F* statistics and p-values shown in brackets above) followed by Tukey's multiple comparisons testing. Statistical significance of multiple comparisons testing is represented by asterisks *$p < 0.05$, ****$p < 0.0001$. †Graphs are provided for visualization purposes only, because non-detectable levels of *UNC13A* cryptic exon- and *SORT1-ex17b*-containing transcripts in some or all control TDP-43^wt and TDP-43^K145R neurons prevented statistical comparisons among groups and therefore no statistical significance was reported.

The online version of this article includes the following figure supplement(s) for figure 3:

**Figure supplement 1.** Development of endogenously encoded acetylation-mimic and acetylation-deficient *TARDBP*-induced pluripotent stem cell (iPSC) lines.

**Figure supplement 2.** Human-induced pluripotent stem cell (iPSC)-derived cortical neuron differentiation.

total *SORT1* transcript in TDP-43^K145Q.12 neurons (***Figure 3D, E***). We conclude that *SORT1+ex1b* is a low-abundance transcript that accumulates in TDP-43^K145Q neurons.

We also examined neurons for the presence of truncated *STMN2* and *UNC13A* cryptic exon mRNA, which are implicated as ALS–FTLD biomarkers and/or risk genes (***Brown et al., 2022***; ***Prudencio et al., 2020***; ***Melamed et al., 2019***; ***Ma et al., 2022***). Strikingly, truncated *STMN2* mRNA was approximately 100-fold higher in both TDP-43^K145Q.12 and TDP-43^K145Q.18 neuron lines, respectively, compared to TDP-43^wt neurons, while no difference from WT was found in TDP-43^K145R neurons. (***Figure 3F***). The unperturbed *STMN2* splice variant (*STMN2* WT) was reduced about 50% in both acetylation-mimic TDP-43^K145Q lines but unchanged in TDP-43^K145R.2 neurons (***Figure 3G***). We also noted a subtle increase in *STMN2* WT mRNA in TDP-43^K145R neurons, suggesting the fully deacetylated state, achieved by acetylation-deficient mimics, may augment *STMN2* levels. RT-qPCR analysis of cryptic exon-included *UNC13A* splice variant (*UNC13A* cryptic exon) failed to detect the abnormal splice product in TDP-43^wt and TDP-43^K145R.2 lines, with only low levels observed in some TDP-43^K145R.12 samples. In contrast, both TDP-43^K145Q lines dramatically accumulated the *UNC13A* cryptic exon containing mRNA (***Figure 3H***). The mRNA levels of the typical splice variant (*UNC13A* WT) were similar across all five lines (***Figure 3I***), indicating that the lack of detectable *UNC13A* cryptic exon mRNA in WT and acetylation-deficient neuron lines was not a result of reduced *UNC13A* expression, but rather is likely specific to a lack of cryptic splicing events. Taken together, these results demonstrate that endogenously expressed acetylation-mimic TDP-43^K145Q alters TDP-43 localization and conformation in response to stress and impairs splicing in a disease-relevant manner in human iPSC-derived neurons.

## TDP-43 acetylation-mimic mice develop age-dependent cognitive and behavioral defects

To evaluate TDP-43 acetylation-mimic mice, we aged the animals and performed an extensive battery of behavioral analysis to assess cognitive and motor function, which reflect impairments that are commonly found in patients with TDP-43 proteinopathies (***Geser et al., 2010***). To be sufficiently powered to detect small differences in behavioral phenotypes, we initially focused on TDP-43^wt and homozygous TDP-43^KQ/KQ mice. Moreover, since there were no significant differences between males and females in any behavioral analyses described below, we pooled both sexes into either WT or TDP-43^KQ/KQ groups. At 12 months old, TDP-43^KQ/KQ mice showed significant reduction of body weight compared to WT littermates, and this difference was maintained until end point analysis at 18 months old (***Figure 4A***). Evaluation of exploratory activity and locomotion in an open-field test demonstrated that TDP-43^KQ/KQ mice spend significantly more time in the center region (***Figure 4B***), with no differences in total distance traveled (***Figure 4—figure supplement 3A***), indicative of decreased anxiety-like behavior (***Carola et al., 2002***; ***Seibenhener and Wooten, 2015***). Acoustic startle testing revealed impaired prepulse inhibition (PPI) in TDP-43^KQ/KQ mice at 12 months old (***Figure 4—figure supplement 2A***), indicative of deficits in sensorimotor gating, a form of inhibitory behavioral control (***Gómez-Nieto et al., 2020***; ***Mena et al., 2016***), which is a phenomenon that can be observed in early dementia (***Ueki et al., 2006***). In 18-month-old animals, evaluation of acoustic startle response and PPI was confounded by hearing impairment (***Figure 4—figure supplement 2B***), however the altered activity reflected by increased time in the center region of an open-field test was maintained

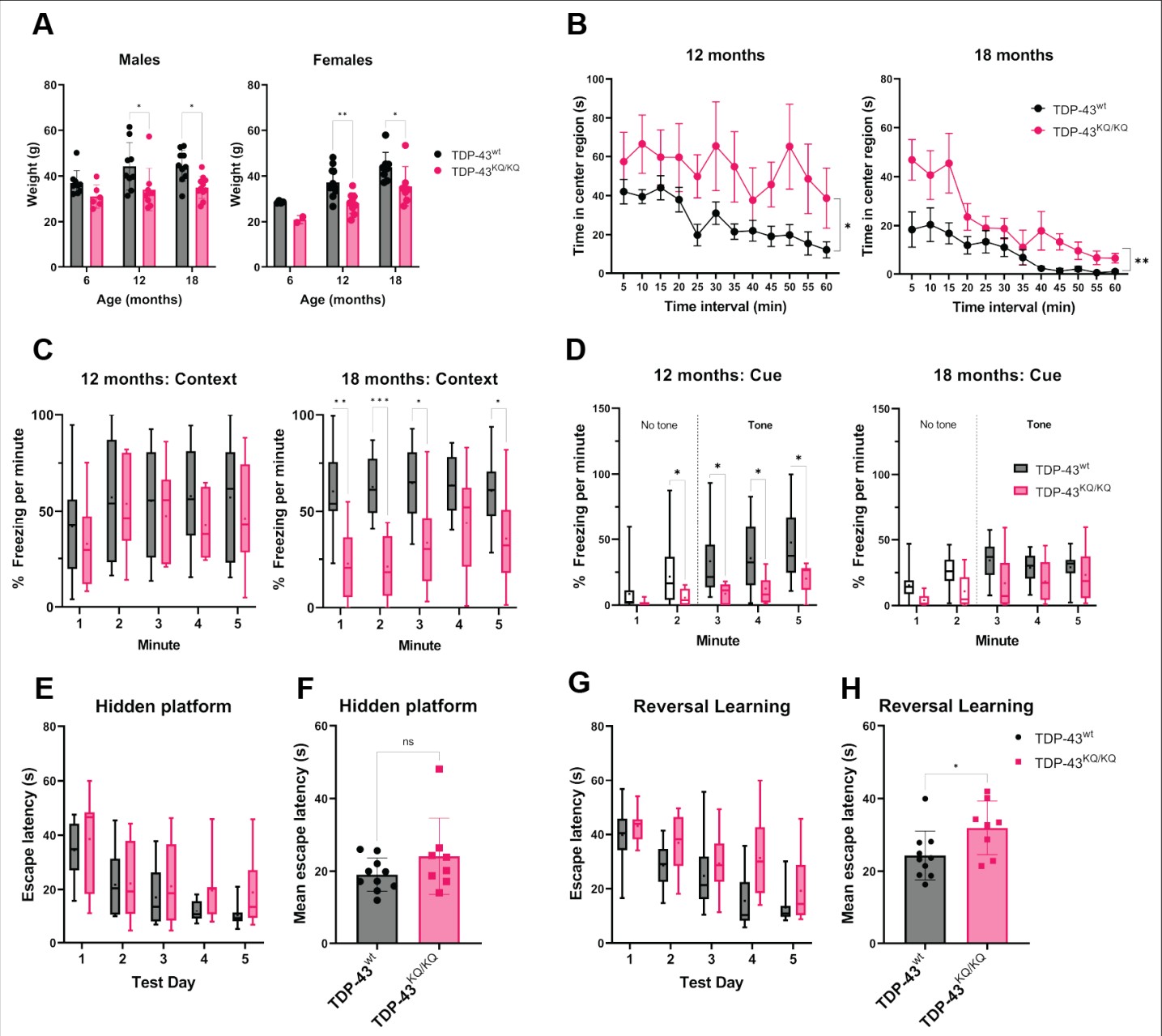

**Figure 4.** TDP-43[KQ/KQ] mice develop age-dependent cognitive and behavioral defects. (**A**) Body weight of TDP-43[wt] and TDP-43[KQ/KQ] mice in males [left panel; $F_{(1,48)}$ = 17.52, p = 0.0001] and females [right panel; $F_{(1,36)}$ = 14.17, p = 0.0006] at different ages. (**B**) Quantification of time in the center region of an open field in mice at 12 months [left panel; $F_{(1,20)}$ = 6.118, p = 0.0225] and 18 months [right panel; $F_{(1,17)}$ = 9.622, p = 0.0065). (**C**) Quantification of time spent frozen (immobile) following context-dependent conditioned fear testing at 18 months old [right panel; $F_{(1,17)}$ = 5.402, p = 0.0328]. (**D**) Quantification of time spent frozen following cue-dependent conditioned fear testing at 12 months old [left panel; $F_{(1,20)}$ = 6.136, p = 0.0223] and 18 months old. Filled bars represent presence of auditory cue (tone) and period of statistical analysis. Morris Water Maze (MWM) analysis displaying time to find a hidden platform (escape latency), quantified as daily trial means per animal (**E**) or average across all days (**F**). Quantification of escape latencies during MWM reversal learning trials in daily trials (**G**) [$F_{(1,16)}$ = 5.273, p = 0.0355] and comparing averages per mouse across all days (**H**). Bar and scatter plots shown as mean ± standard deviation (SD). Box and whiskers show line at median, '+' at mean, and whiskers run min to max. (**A**) Two-way analysis of variance (ANOVA) followed by Šídák's multiple comparisons test. (B–D, E, G) Two-way repeated measures ANOVA followed by Holm–Šídák's multiple comparisons tests; F statistics and p-values in legend represent main effect of genotype. (**F, H**) Unpaired Student's t-test. Sample sizes as follows unless otherwise indicated: 12-month TDP-43[wt] n = 15; 12-month TDP-43[KQ/KQ] n = 7; 1-month TDP-43[wt] n = 10; 18-month TDP-43[KQ/KQ] n = 9; one 18-month TDP-43[KQ/KQ] extreme outlier removed for MWM analysis (**E, F**). Statistical significance is represented by asterisks *p < 0.05, **p < 0.01, ***p < 0.001; ns = not significant. Further statistical information is located in *Figure 4—source data 1* file.

The online version of this article includes the following source data and figure supplement(s) for figure 4:

*Figure 4 continued on next page*

*Figure 4 continued*

**Source data 1.** Complete statistical results and information for quantitative elements shown in *Figure 4*.

**Figure supplement 1.** Heterozygous TDP-43[wt/KQ] animals exhibit mild cognitive and behavioral deficits at levels intermediate to TDP-43[wt] and TDP-43[KQ/KQ] mice.

**Figure supplement 1—source data 1.** Complete statistical results and information for quantitative elements shown in *Figure 4—figure supplement 1*.

**Figure supplement 2.** Altered sensorimotor gating and impaired prepulse inhibition (PPI) in TDP-43[KQ/KQ] mice.

**Figure supplement 2—source data 1.** Complete statistical results and information for quantitative elements shown in *Figure 4—figure supplement 2*.

**Figure supplement 3.** Lack of overt motor deficits in TDP-43[KQ/KQ] mice up to 18 months of age.

**Figure supplement 3—source data 1.** Complete statistical results and information for quantitative elements shown in *Figure 4—figure supplement 3*.

at this age (*Figure 4B*). Thus, consistent patterns of behavioral disinhibition and reduced anxiety-like behavior were apparent in TDP-43[KQ/KQ] mice over time.

We next performed contextual and cue-dependent fear conditioning as an index of hippocampal and cortical function (*Chen et al., 1996*; *Curzon et al., 2009*; *Phillips and LeDoux, 1992*; *Kim and Jung, 2006*; *Marschner et al., 2008*). Context-dependent fear testing revealed reduced freezing times in TDP-43[KQ/KQ] mice, with trends observed at 12 months old and significant deficits appearing at 18 months (*Figure 4C*). Similarly, auditory cue-dependent fear testing revealed significant impairments in associative cue learning in TDP-43[KQ/KQ] mice at 12 months of age (*Figure 4D*), a behavior thought to be mediated by the amygdala and higher-order cortical regions important in inhibitory control (*Sierra-Mercado et al., 2011*). As mentioned above, general auditory defects in both genotypes at 18 months of age confounded interpretations of any cue-dependent learning deficits at this advanced age (*Figure 4D*). Morris Water Maze testing was used to evaluate swimming ability and spatial learning (*Vorhees and Williams, 2006*), which showed equivalent swim speeds, suggesting no motor impairments in TDP-43[KQ/KQ] mice at 18 months of age (*Figure 4—figure supplement 2D*). While assessment of spatial learning showed a trend toward impaired acquisition learning (*Figure 4E, F*), we observed more prominent defects in reversal learning after moving the location of the platform, as determined by significant delays in escape latency. Our findings support deficits in cognitive flexibility in TDP-43[KQ/KQ] mice (*Figure 4G, H*; *Nicholls et al., 2008*). We also analyzed cognitive impairments in heterozygous TDP-43[wt/KQ] mice, which showed a mild intermediate phenotype (*Figure 4—figure supplement 1*).

To determine any ALS-like motor phenotypes, we assessed motor function and were surprised to find no overt signs of motor impairment in TDP-43[KQ/KQ] mice at 18 months of age. TDP-43[KQ/KQ] mice performed similarly to WT littermates as assessed by rotarod testing (*Figure 4—figure supplement 3C, D*) and in grip strength as measured using digital force meters (*Figure 4—figure supplement 3E*). Moreover, there were no differences in swim speed or distance traveled in an open field at any age tested (*Figure 4—figure supplement 3A, B*). We note that we cannot currently exclude the possibility of more subtle motor phenotypes or the emergence of motor defects in aged mice beyond 18 months old. The preferential deficits in learning and behavioral control support an age-dependent FTLD- or dementia-like phenotype in acetylation-mimic TDP-43[KQ/KQ] mice.

## TDP-43[KQ/KQ] mice show hallmarks of TDP-43 dysfunction in the absence of neurodegeneration

Given the preferential cognitive phenotype observed in TDP-43[KQ/KQ] mice (*Figure 4*), we focused on the neocortex and hippocampus of aged animals. We examined neuronal density in the neocortex of 12- and 18-month-old TDP-43[wt] and TDP-43[KQ/KQ] mice by immunofluorescence, widefield microscopy, and quantitative image analysis of the NeuN-positive neuronal cell density and area occupied by NeuN-positive cells in the neocortex. We did not observe any evidence of neuron loss in the neocortex of TDP-43[KQ/KQ] mice at 12 or 18 months of age (*Figure 5A–C*). Similarly, we observed no astrogliosis (GFAP intensity and area occupied by GFAP+ cells) or microgliosis (Iba1 intensity and area occupied by Iba1+ cells) in the neocortex or hippocampus (*Figure 5—figure supplement 1*).

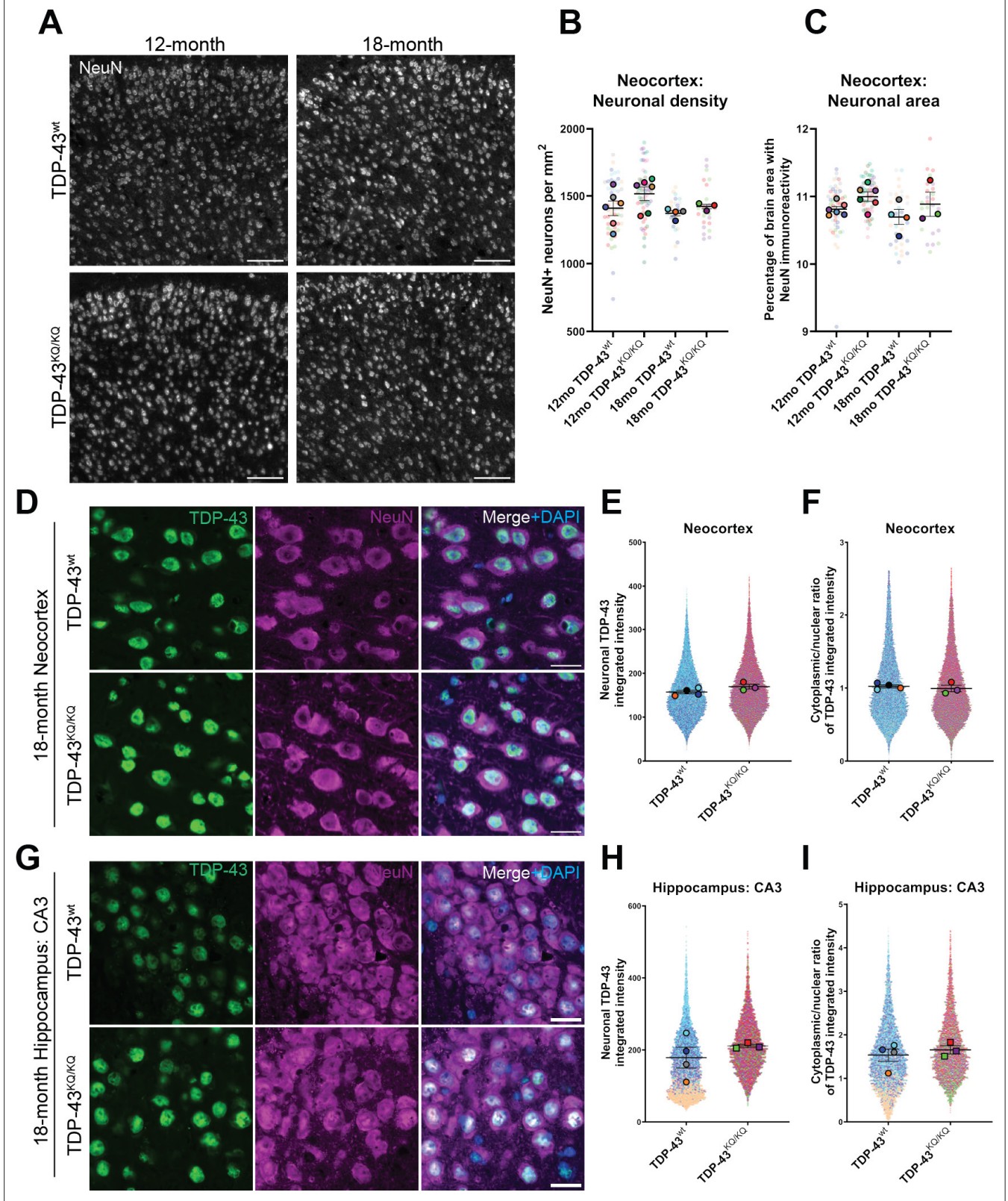

**Figure 5.** No evidence of neuronal loss in aged TDP-43^KQ/KQ mice with retention of predominantly nuclear TDP-43. (**A**) Representative confocal immunofluorescence images of NeuN+ neurons in the neocortex of 12- and 18-month TDP-43^wt or TDP-43^KQ/KQ mice. (**B**) Quantification of NeuN+ neurons per square millimeter (mm) in TDP-43^wt and TDP-43^KQ/KQ mice neocortex. (**C**) Quantification of the percent of neocortex area that is occupied by cells with NeuN immunoreactivity. Representative confocal images of neocortex (**D**) and hippocampus (**G**) sections from 18-month-old TDP-43^wt

*Figure 5 continued on next page*

*Figure 5 continued*

and TDP-43$^{KQ/KQ}$ mice immunolabeled with TDP-43 (green), NeuN (magenta), and DAPI (blue). (**E, H**) Quantification of TDP-43 fluorescence intensity within NeuN+ neurons in the neocortex (**F**) or hippocampal CA3 region (**H**). (**F, I**) Quantification of the nuclear:cytoplasmic ratio of TDP-43 fluorescence intensity within NeuN+ neurons in the neocortex (**F**) and hippocampus (**I**). Scale bars = 100 µm (**A**), 20 µm (**D, G**). SuperPlots (*Lord et al., 2020*) show average value per animal in solid color bordered symbol (as mean ± standard error of the mean [SEM]) over top of semi-transparent individual values from each animal. One color represents one animal. Individual datapoints in (**B**) and (**C**) represent density in a single field of view, with 10–16 fields across 4 brain sections per animal, *n* = 3–4 mice per genotype of 18-month-old animals and *n* = 6 mice per genotype of 12-month-old animals. Individual datapoints in E, F and H, I represent values from a single neuron within one field of view, with 1000–5000 neurons per animal across 12–16 fields from 4 (neocortex) or 2 (hippocampus) brain sections of *n* = 3–4 mice per genotype of 18-month-old animals. Statistical analyses performed using linear mixed-effect models. Omission of asterisks indicates no statistical significance.

The online version of this article includes the following figure supplement(s) for figure 5:

**Figure supplement 1.** No evidence of astrogliosis or microgliosis in the hippocampus or neocortex of 18-month-old TDP-43$^{KQ/KQ}$ mice.

We next evaluated TDP-43 aggregation and cytoplasmic mislocalization in 18-month-old mice, a time at which cognitive deficits were most pronounced (*de Boer et al., 2021*; *Neumann, 2009*). We initially used confocal microscopy, and automated quantitative image analysis to assess TDP-43 morphology and localization in the neocortex and hippocampus, which did not reveal nuclear or cytoplasmic TDP-43-positive inclusions in any brain region or genotype (*Figure 5D, G*). In addition to searching for aggregate pathology, we also evaluated TDP-43 abundance, as measured by fluorescence intensity, in NeuN-positive neurons, as TDP-43 expression is altered in FTLD-TDP and other TDP-43 proteinopathies (*Budini and Buratti, 2011*; *Mishra et al., 2007*; *Gitcho et al., 2009*; *Chen-Plotkin et al., 2008*), and visualized the single-cell resolution data using SuperPlots (*Lord et al., 2020*). We observed trends toward elevated TDP-43 protein levels in the neocortex and the CA3 region of the hippocampus (*Figure 5E, F*) and potential subtle increases in the cytoplasmic to nuclear TDP-43 ratio within CA3 neurons (*Figure 5G, J*), however these effects were variable and not statistically significant. Given the technical challenge of accurate image segmentation at the subcellular level within tissue samples, we sought more sensitive methods to assess TDP-43 pathology in TDP-43$^{KQ/KQ}$ mice.

We turned to an alternative biochemical approach of sequentially fractionating isolated hippocampus and neocortex tissue to generate soluble (Radioimmunoprecipitation assay (RIPA) buffer-extracted) and insoluble (Urea-extracted) protein fractions. TDP-43$^{KQ/KQ}$ mouse neocortex harbored insoluble phosphorylated TDP-43 at the disease-associated Ser409/410 locus (*Neumann et al., 2009*) (p409/410) at 12 months of age, which was even more prominent at 18 months (*Figure 6A, C*), a timepoint at which TDP-43$^{KQ/KQ}$ mice showed behavioral and cognitive defects (*Figure 4*). Notably, p409/410 abundance was minimal in the hippocampus of 12-month-old TDP-43$^{KQ/KQ}$ mice but increased by 18 months (*Figure 6D, F*), coinciding with the onset of hippocampal-mediated learning deficits (*Figure 4C*). Though p409/410 was elevated in TDP-43$^{KQ/KQ}$ mice, we were surprised to find that the total insoluble TDP-43 pool was not altered, suggesting that increased TDP-43 phosphorylation may precede overt conversion from soluble to insoluble TDP-43.

We also examined soluble TDP-43 levels, as acetylation-induced loss of function may result in auto-regulatory feedback that increases the production of TDP-43 (*Budini and Buratti, 2011*). The soluble TDP-43 pool was significantly increased in TDP-43$^{KQ/KQ}$ at 12 months in the neocortex and at 18 months in both neocortex (*Figure 6A, B*) and hippocampus (*Figure 6D, E*), an indicator of autoregulated TDP-43 protein levels. Biochemical analysis of TDP-43 solubility and phosphorylation in heterozygous TDP-43$^{wt/KQ}$ animals revealed no significant differences in soluble TDP-43 or insoluble p409/410 levels compared to TDP-43$^{wt}$ mice (*Figure 6—figure supplement 1A–C*, *Figure 6—figure supplement 2A–C*). To assess TDP-43 mislocalization using a biochemical approach, we performed subcellular fractionation to isolate nuclear and cytoplasmic proteins from the neocortex and hippocampus and found a striking increase in cytoplasmic TDP-43 in TDP-43$^{KQ/KQ}$ mice at both 12 and 18 months of age (*Figure 6G–L*). Similar to the intermediate levels of cognitive decline in heterozygous TDP-43$^{wt/KQ}$ mice, we observed trends toward increased cytoplasmic TDP-43 in TDP-43$^{wt/KQ}$ mice, particularly at 18 months old (*Figure 6—figure supplement 1D–I*, *Figure 6—figure supplement 2D–I*).

Finally, we evaluated whether TDP-43$^{KQ/KQ}$ mice show TDP-43 dysfunction in the spinal cord. Surprisingly, TDP-43$^{KQ/KQ}$ mice harbored insoluble p(409/410)-TDP-43 (*Figure 6—figure supplement 3A, C*) and moderate increases in cytoplasmic TDP-43 mislocalization (*Figure 6—figure supplement*

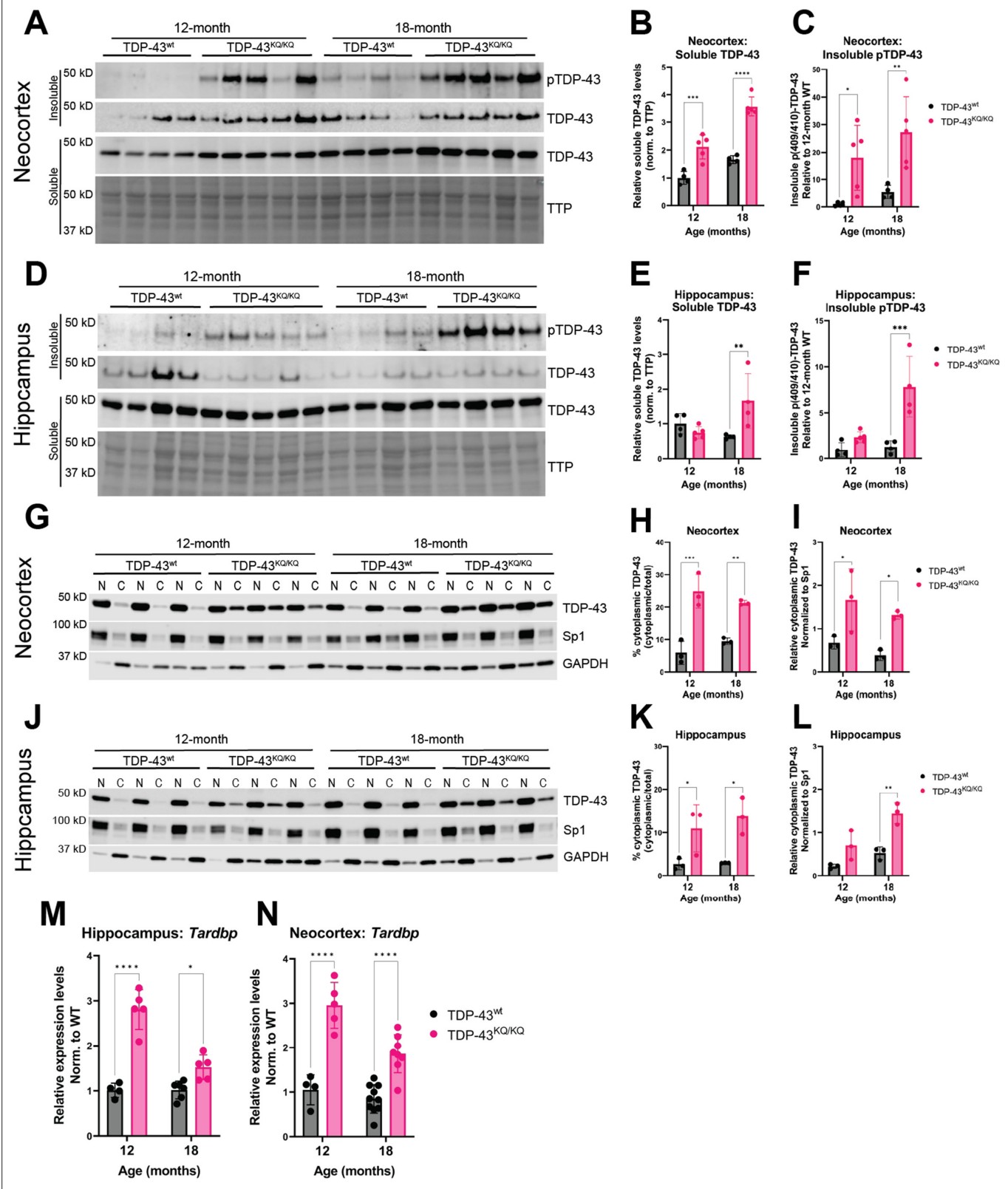

**Figure 6.** Hyperphosphorylated and mislocalized TDP-43 protein and autoregulated induction of the *Tardbp* transcript is found in the neocortex and hippocampus of aged TDP-43[KQ/KQ] mice. (**A–F**) Western blot images comparing soluble and insoluble protein fractions in neocortex (**A**) and hippocampus (**D**) from TDP-43[wt] and TDP-43[KQ/KQ] mice at 12 and 18 months of age. Quantification of soluble TDP-43 protein levels in neocortex and hippocampus tissue relative to total transferred protein (TTP) (neocortex (**B**) $F(1,14)$ = 97.67, p < 0.0001; hippocampus (**E**) main effect of genotype

*Figure 6 continued on next page*

*Figure 6 continued*

$F_{(1,13)} = 3.699$, p = 0.0766 and main effect of genotype × age interaction $F_{(1,13)} = 10.21$, p = 0.0070). Quantification of insoluble phosphorylated p(409/410)-TDP-43 levels in neocortex and hippocampus tissue [neocortex (**C**) $F_{(1,14)} = 18.81$, p = 0.0007; hippocampus (**F**) $F_{(1,13)} = 23.07$, p = 0.0003]. (**G–I**) Western blot images following isolation of soluble nuclear and cytoplasmic protein fractions from neocortex (**G**) and hippocampus (**J**) of TDP-43[wt] and TDP-43[KQ/KQ] mice at 12 and 18 months of age. (**J–L**) Quantification of cytoplasmic TDP-43 measured as percent of total TDP-43 in neocortex and hippocampus [neocortex (**H**) $F_{(1,8)} = 69.38$, p < 0.0001; hippocampus (**K**) $F_{(1,8)} = 22.40$, p = 0.0015] and relative cytoplasmic mislocalization of TDP-43 after normalizing to cytoplasmic/nuclear ratio of the nuclear Sp1 protein in neocortex and hippocampus (neocortex (**I**) $F_{(1,8)} = 19.79$, p = 0.0021; hippocampus (**L**) $F_{(1,8)} = 29.94$, p = 0.0006). *Tardbp* expression was analyzed by qPCR in the (**M**) hippocampus [$F_{(1,16)} = 76.86$; p < 0.0001] and (**N**) neocortex [$F_{(1,23)} = 80.74$; p < 0.0001] of TDP-43[KQ/KQ] and TDP-43[wt] mice at 12 and 18 months old. Relative expression (RQ) values in each sample were determined using the Plaff method and *B-actin* and *Pgk1* as housekeeping genes. (**A–F**) n = 4 TDP-43[wt] and n = 5 TDP-43[KQ/KQ] at all ages and regions except 18-month hippocampus n = 4 TDP-43[KQ/KQ]. (**F–L**) n = 4 all ages and regions. (**M**) Neocortex n = 4 12-month TDP-43[wt], n = 5 12-month TDP-43[KQ/KQ], n = 10 18-month TDP-43[wt], n = 9 18-month TDP-43[KQ/KQ]. (**N**) Hippocampus n = 4 12-month TDP-43[wt], n = 5 12-month TDP-43[KQ/KQ], n = 6 18-month TDP-43[wt], n = 6 18-month TDP-43[KQ/KQ]. Data are presented as mean ± standard deviation (SD). Two-way analysis of variance (ANOVA) followed by Šídák's multiple comparisons test. *F* statistics represent the main effect of genotype unless otherwise stated. Statistical significance represented by asterisks, *p < 0.05, **p < 0.01, ***p < 0.001, ****p < 0.0001. Further statistical information is available in *Figure 6—source data 1* file. Figures showing uncropped western blot images are available in *Figure 6—source data 2* files. Full images of western blots are available in *Figure 6—source data 3* files.

The online version of this article includes the following source data and figure supplement(s) for figure 6:

**Source data 1.** Complete statistical results and information for quantitative elements shown in *Figure 6*.

**Source data 2.** Figures showing uncropped western blot images of those presented in *Figure 6A,D,G,J*.

**Source data 3.** Full uncropped images of western blots shown in *Figure 6*.

**Figure supplement 1.** Intermediate levels of TDP-43 hyperphosphorylation and mislocalization in the hippocampus of aged heterozygous TDP-43[wt/KQ] compared to TDP-43[wt] and homozygous TDP-43[KQ/KQ] mice.

**Figure supplement 1—source data 1.** Complete statistical results and information for quantitative elements shown in *Figure 6—figure supplement 1*.

**Figure supplement 1—source data 2.** Figures showing uncropped western blot images of those presented in *Figure 6—figure supplement 1A,D,G*.

**Figure supplement 1—source data 3.** Full uncropped images of western blots shown in *Figure 6—figure supplement 1*.

**Figure supplement 2.** Intermediate levels of TDP-43 hyperphosphorylation and mislocalization in the neocortex of aged heterozygous TDP-43[wt/KQ] compared to TDP-43[wt] and homozygous TDP-43[KQ/KQ] mice.

**Figure supplement 2—source data 1.** Complete statistical results and information for quantitative elements shown in *Figure 6—figure supplement 2*.

**Figure supplement 2—source data 2.** Figures showing uncropped western blot images of those presented in *Figure 6—figure supplement 2A,D,G*.

**Figure supplement 2—source data 3.** Full uncropped images of western blots shown in *Figure 6—figure supplement 2*.

**Figure supplement 3.** Spinal cord tissue from aged TDP-43[KQ/KQ] mice contains insoluble, hyperphosphorylated, and mislocalized TDP-43.

**Figure supplement 3—source data 1.** Complete statistical results and information for quantitative elements shown in *Figure 6—figure supplement 3*.

**Figure supplement 3—source data 2.** Figures showing uncropped western blot images of those presented in *Figure 6—figure supplement 3A,D*.

**Figure supplement 3—source data 3.** Full uncropped images of western blots shown in *Figure 6—figure supplement 3*.

3D–F) in the spinal cord at 18 months old, despite the lack of any overt motor deficits in TDP-43[KQ/KQ] animals (*Figure 4—figure supplement 2*). We thus found that a similar biochemical signature of TDP-43 abnormalities is present in the spinal cord, yet does not correlate with motor deficits in 12- to 18-month-old mice. Our findings suggest TDP-43 phosphorylation and mislocalization occur as early events, prior to the formation of mature TDP-43 inclusion pathology and may drive the progressive FTLD-TDP-like phenotype in TDP-43[KQ/KQ] mice.

## Disease-linked transcriptomic and splicing defects are prevalent in acetylation-mimic TDP-43[KQ/KQ] mice

TDP-43 acetylation drives RNA dissociation and loss of TDP-43 function (*Cohen et al., 2015*), implying that reduced RNA-binding capacity may impact transcriptional regulation and mRNA splicing (*Duan et al., 2022*; *Sternburg et al., 2022*; *Hallegger et al., 2021*). We performed total RNA sequencing to determine how mimicking TDP-43 acetylation affects RNA profiles in vivo. We examined the neocortex and hippocampus of 18-month-old TDP-43[wt] or TDP-43[KQ/KQ] mice, representing the regions and time-points with the most striking biochemical and behavioral and abnormalities and identified nearly 400

differentially expressed genes (DEGs) in each brain region in TDP-43[KQ/KQ] mice compared to TDP-43[wt], after correcting for underlying batch effects. As expected by acetylation-induced loss of TDP-43 function and subsequent autoregulation, the *Tardbp* transcript was increased (*Supplementary file 1a, b*). Follow-up RT-qPCR analysis confirmed a two- to threefold increase in *Tardbp* expression in the neocortex and hippocampus (*Figure 6M, N*), as well as the spinal cord (*Figure 7—figure supplement 1A*), which correlated with increased TDP-43 protein levels (*Figure 6B, E*; *Figure 6—figure supplement 3A–C*).

We then clustered the DEGs based on their up- or downregulation and the brain region affected (*Figure 6*), which revealed largely similar patterns of transcriptional alterations, with some distinct differences (*Figure 7*, *Figure 7—figure supplement 2*, *Supplementary file 1a, b*). To investigate the potential biological implications of the altered transcriptome, we performed Gene Ontology (GO) term enrichment analyses on DEGs identified in each of the six clusters (*Supplementary file 1c*). In both brain regions, the most highly downregulated genes were involved in developmental processes, including many related to CNS development and maintenance, such as neurogenesis (e.g., *Sema5b*, *Rnd2*, *Brinp1*), gliogenesis (e.g., *Tlr2*, *Olig2*, *Sox10*), and myelination (e.g., *Nkx2-2*, *Nkx6-2*, *Sox10*). *Sema5b* was the most dramatically reduced transcript in the hippocampus, and the third most highly reduced in the cortex (*Supplementary file 1a, b*) with a twofold reduction in expression in TDP-43[KQ/KQ] mice. Downregulated genes in both the hippocampus and neocortex were also enriched for terms related to synapse homeostasis and transmembrane signaling, however the dysregulated pathways were distinct. Genes related to GABAergic synapses (e.g., *Gad1*, *Gad2*, *Abat*, *Gnb5*) were selectively downregulated in the neocortex, while trans-synaptic signaling and ion transport mechanisms (e.g., *Homer3*, *Camk4a*, *Nsmf*, *Cnih2*) were decreased in the hippocampus.

In contrast, there was significant upregulation of cellular stress response genes (*Sesn1*, *Nrros*, *Plat*, *Klf15*) and many apoptotic regulators (e.g., *Trp53inp1*, *Pmaip1*, *Bcl2l1*, *Plekhf1*) in both the hippocampus and neocortex, along with an over-representation of GO terms related to metabolism, localization, and cell adhesion (*Figure 7*, *Supplementary file 1c*). Several pathways were uniquely altered including coagulation and complement cascades, which were only upregulated in the hippocampus (e.g., *F3*, *Plat*, *Cd59a*). Interestingly, while genes associated with trans-synaptic signaling were decreased in the hippocampus, another set of genes involved in this same pathway was upregulated in the cortex (e.g., *Syt7*, *Synpo*, *Nptx1*, *Spg11*) (*Supplementary file 1c*).

Given the cognitive defects in TDP-43[KQ/KQ] mice, we sought to draw parallels between the TDP-43[KQ/KQ] mouse and the human FTLD-TDP transcriptome. Comparison of the DEGs in TDP-43[KQ/KQ] mice to the mouse orthologs of those found in FTLD-TDP frontal or temporal cortex tissue (*Hasan et al., 2022*) revealed marked overlap between our mouse and their human datasets (*Figure 7*, see 'FTLD-TDP Cortex' bar), particularly in the hippocampus, as measured by statistical enrichment analyses that found over-representation of FTLD-TDP DEGs in our TDP-43[KQ/KQ] dataset (*Supplementary file 1d*; p = 0.0003 'Hippocampus Down' vs Downregulated in FTLD-TDP frontal cortex; p = 0.0014 'Hippocampus Down' vs Downregulated in FTLD-TDP temporal cortex; p = 0.0048 'Hippocampus Up' vs Upregulated in FTLD-TDP frontal cortex). A similar alignment comparing DEGs from TDP-43[KQ/KQ] mice to those following TDP-43 knockdown in mouse striatum also identified commonly altered genes, particularly in the TDP-43[KQ/KQ] downregulated gene sets (*Figure 7*, see 'TDP-43 knockdown' bar; *Supplementary file 1d*; p = 0.019 'Hippocampus Down' vs Downregulated in TDP-43-KD; p = 0.046 'Cortex Down + Hippocampus Down' vs Upregulated in TDP-43-KD). Taken together, we identified distinct FTLD-TDP signatures including altered synaptic gene expression and stress response signaling that reflect acetylation-induced TDP-43 dysfunction.

Alternative splicing defects, particularly impaired repression of cryptic exons, due to TDP-43 dysregulation are strongly implicated in FTLD and ALS pathogenesis (*Melamed et al., 2019*; *Ma et al., 2022*; *Ling et al., 2015*; *Arnold et al., 2013*; *Prudencio et al., 2020*; *Humphrey et al., 2017*). In line with our findings above that TDP-43[KQ/KQ] mouse primary cortical neurons in vitro show splicing deficits (*Figure 2E, F*), we identified widespread splicing alterations in vivo. Analysis of TDP-43[KQ/KQ] neocortex identified 289 differentially spliced genes (DSGs), with 81.7% of loci containing at least one cryptic splice junction, and 29.8% containing two cryptic splice junctions (*Supplementary file 1e*). In the hippocampus, we found 126 DSGs, 77.0% of which contain a cryptic splice junction and 41.3% that are formed by two cryptic splice sites (*Supplementary file 1f*). The alternative splicing events were relatively consistent between brain regions, as over 70% of the DSGs identified in the hippocampus

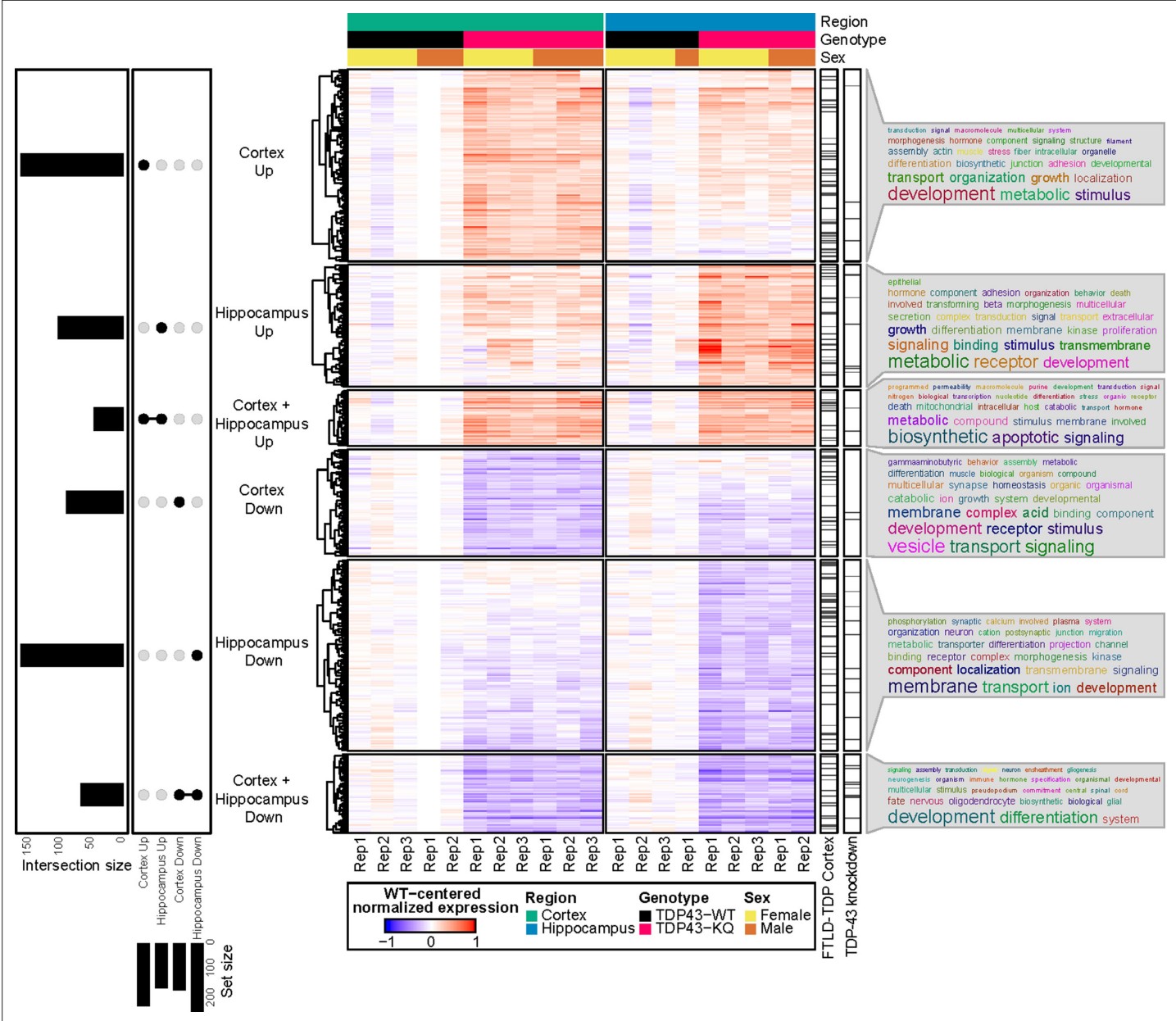

**Figure 7.** RNA sequencing of aged TDP-43$^{KQ/KQ}$ mouse brain reveals dysregulation of neurodegeneration and FTLD-TDP-associated pathways. Genes differentially expressed between TDP-43$^{KQ/KQ}$ (TDP43-KQ) and TDP-43$^{wt}$ (TDP43-WT) cortex and hippocampus were compared and their intersections were defined as six gene groups, as represented by an UpSet plot (left). Normalized gene expression values, centered around the mean of TDP-43$^{wt}$ samples for each brain region, were then hierarchically clustered and plotted as a heatmap (center, red indicates expression higher than that of WT, blue indicates expression lower than WT). Genes that were significantly altered in human FTLD-TDP temporal and/or frontal cortex (**Hasan et al., 2022**, 'FTLD-TDP Cortex' bar), or striatal mouse brain TDP-43-knockdown (**Polymenidou et al., 2011**, 'TDP-43 knockdown' bar) were demarcated with tick marks. Each gene group was assessed for over-enrichment of Gene Ontology terms and these results were summarized as word clouds (right). Neocortex $n = 5$ TDP-43$^{wt}$, $n = 6$ TDP-43$^{KQ/KQ}$; hippocampus $n = 4$ TDP-43$^{wt}$, $n = 5$ TDP-43$^{KQ/KQ}$.

The online version of this article includes the following source data and figure supplement(s) for figure 7:

**Figure supplement 1.** Spinal cord tissue from TDP-43$^{KQ/KQ}$ mice shows evidence of altered TDP-43 autoregulation and aberrant splicing of *Sort1* compared to TDP-43$^{wt}$.

**Figure supplement 1—source data 1.** Complete statistical results and information for quantitative elements shown in *Figure 7—figure supplement 1*.

**Figure supplement 2.** Significant correlation between differentially expressed genes (DEGs) in the neocortex and hippocampus of TDP-43$^{KQ/KQ}$ mice.

were also present in the cortex. Among the most significant DSGs identified were known TDP-43 splicing targets (e.g., *Kcnip2, Pdp1, Poldipp3, Ppfibp1, Dnajc5, Tmem2, Sort1*) (*Fratta et al., 2018*; *Arnold et al., 2013*; *Polymenidou et al., 2011*), transcripts associated with particular neurodegenerative diseases (e.g., *Mapt, Atxn1, Lrrk2*) (*Trabzuni et al., 2012*; *Park et al., 2016*; *Giesert et al., 2013*; *Lacognata et al., 2015*; *Rosas et al., 2020*; *Manek et al., 2020*; *Banfi et al., 1994*), and also many robustly altered transcripts that are poorly characterized but linked to neurodegeneration (e.g., *Nrxn3, Nos1, Arfgef2, Arhgap10, Lrp8, Smarca4, Rims2*). Of all identified DSGs, the most substantially altered transcript was *Sort1,* encoding the Sortilin-1 (Sort1) protein. We noticed that exclusion of a *Sort1* 3' exon was reduced by 55.9% in TDP-43$^{KQ/KQ}$ cortex and by 57.0% in the hippocampus (*Figure 8A*), such that the aberrantly spliced *Sort1* transcript was the predominant variant detected, comprising nearly 80% of all *Sort1* transcripts, aligning with our findings from TDP-43$^{KQ/KQ}$ primary mouse cortical neurons.

SORT1, the human homolog of mouse Sort1, is a highly expressed neurotrophic factor receptor that binds progranulin (PGRN) and regulates endosomal/lysosomal function through a pathway that is genetically linked to FTLD-TDP (*Carlo et al., 2014*; *Hu et al., 2010*; *Xu et al., 2019*; *Pallesen and Vaegter, 2012*; *Mohagheghi et al., 2016*). Because TDP-43 depletion results in *SORT1* exon 17b inclusion, this leads to the production of a soluble and putatively toxic SORT1 variant that is increased in FTLD-TDP patients (*Prudencio et al., 2012*; *Tann et al., 2019*; *Polymenidou et al., 2011*). To confirm that the altered 3' splicing in our sequencing data was indeed *Sort1* exon 17b inclusion, we performed qPCR using primers specific for the mouse *Sort1+ex17b* transcript or the appropriately spliced variant (*Sort1-WT*), and a primer pair that recognizes all *Sort1* variants (*Sort1* total) on tissues isolated from TDP-43$^{KQ/KQ}$ or WT mice (*Prudencio et al., 2012*). We found the ratio of aberrant *Sort1-ex17b* to appropriately spliced *Sort1-WT* was increased approximately 10-fold in the neocortex (*Figure 8B*) and hippocampus (*Figure 8C*) of TDP-43$^{KQ/KQ}$ mice at both 12 and 18 months of age, which corroborates our RNA sequencing analysis showing that the exon-included *Sort1+ex17b* variant predominates in these tissues. The total level of *Sort1* transcript in vivo varied with age, with slightly increased levels at 12 months and reduced levels at 18 months of age, suggestive of a negative feedback mechanism regulating *Sort1* expression. Spinal cord tissue revealed similar levels of abnormal *Sort1* splicing in TDP-43$^{KQ/KQ}$ animals at 12 and 18 months of age, however total *Sort1* mRNA expression did not significantly change with age (*Figure 7—figure supplement 1B, C*).

To determine whether the *Sort1+ex17b* transcript in TDP-43$^{KQ/KQ}$ mice leads to the generation of a distinct Sort1 protein product, we immunoblotted hippocampus and cortex homogenates and identified a higher molecular weight Sort1 variant in TDP-43$^{KQ/KQ}$ compared to WT mice (*Figure 8D, E*), consistent with *Sort1* exon 17b inclusion. The abnormal Sort1 protein showed decreased steady-state stability in TDP-43$^{KQ/KQ}$ mice, as suggested by reduced protein levels in acetylation-mimic animals, compared to controls (*Figure 8F, G*). These data are strongly suggestive of failure to repress exon inclusion in TDP-43$^{KQ/KQ}$ mice, leading to an altered Sort1-progranulin axis and supporting an aberrant splicing profile that resembles alterations seen in human FTLD-TDP.

## Discussion

Here, we developed novel neuronal and mouse models of RNA-binding deficient acetylation-mimic TDP-43 (TDP-43$^{K145Q}$). Using a combination of molecular, cellular, and transcriptomic readouts combined with in vivo behavioral assays, we provide evidence that aberrant TDP-43$^{K145Q}$ recapitulates key hallmarks of TDP-43 proteinopathies, including altered TDP-43 solubility and localization, impaired autoregulation, and abnormal splicing function. Taken together, our findings suggest that TDP-43 acetylation-induced dysfunction likely contributes to disease progression.

Our prior mass spectrometry analysis originally identified acetylated lysines K145 and K192 (*Cohen et al., 2015*), though a more recent study identified residues K84 (within the NLS) and K136 (within RRM1) as additional lysine residues of interest (*Garcia Morato et al., 2022*). Any normal physiological or cellular role for TDP-43 acetylation remains unclear. However, given its impact on RNA binding, one can imagine TDP-43 acetylation may fine-tune TDP-43's normal nuclear and cytoplasmic functions, but becomes corrupted due to aging and/or stress exposure (*Ebert et al., 2022*). It is likely that acetylation within the RRMs generally impairs TDP-43 RNA-binding function, as multiple studies have shown that acetylation-mimic TDP-43$^{K136Q}$ or TDP-43$^{K145Q}$ variants phenocopy one another, with both showing nuclear foci formation and reduced affinity for target RNAs. Therefore, deficiency in RNA binding may

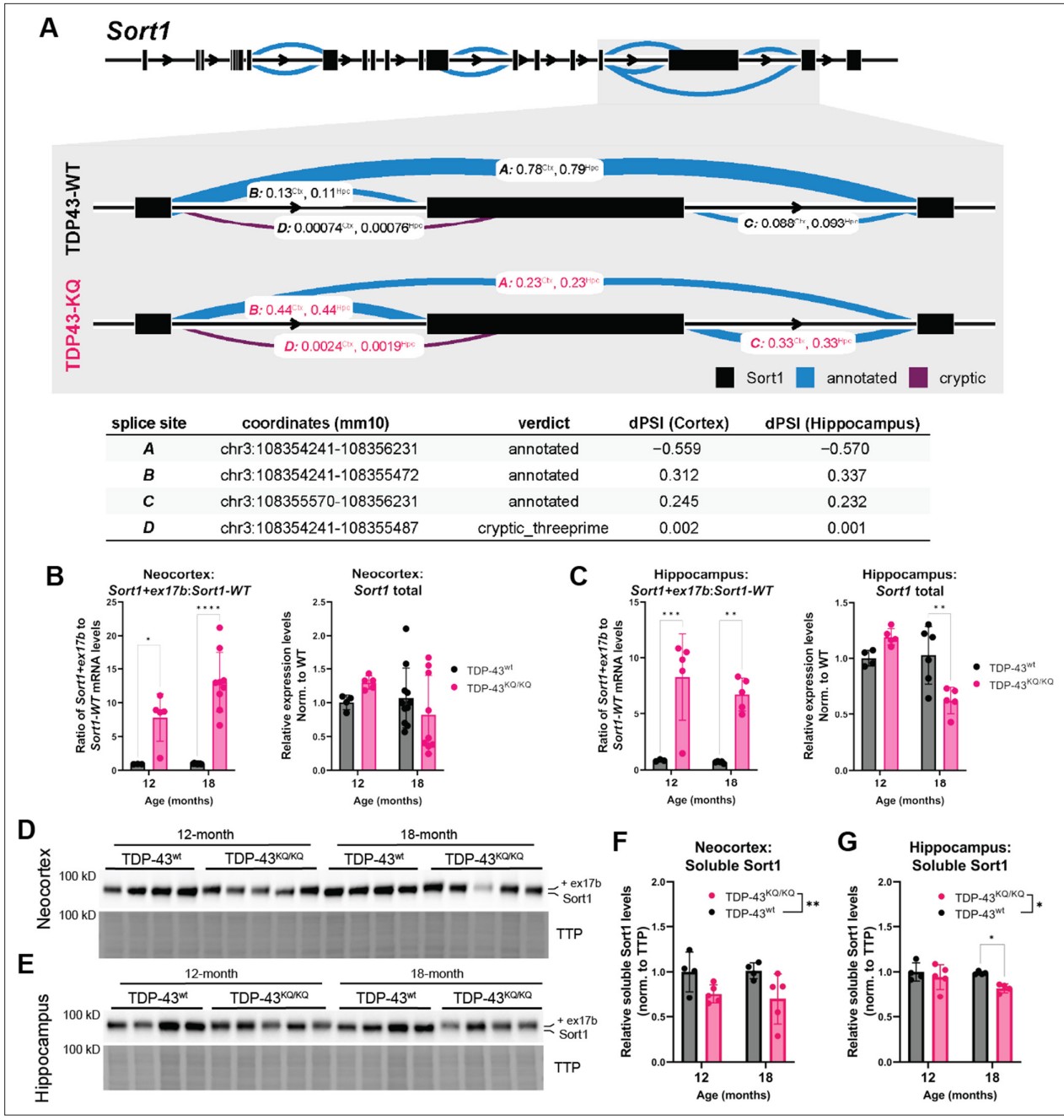

| splice site | coordinates (mm10) | verdict | dPSI (Cortex) | dPSI (Hippocampus) |
|---|---|---|---|---|
| A | chr3:108354241-108356231 | annotated | −0.559 | −0.570 |
| B | chr3:108354241-108355472 | annotated | 0.312 | 0.337 |
| C | chr3:108355570-108356231 | annotated | 0.245 | 0.232 |
| D | chr3:108354241-108355487 | cryptic_threeprime | 0.002 | 0.001 |

**Figure 8.** Dysfunctional splicing regulation of the *Sort1* transcript in TDP-43$^{KQ/KQ}$ mice. (**A**) Differential splicing analysis of TDP-43$^{wt}$ and TDP-43$^{KQ/KQ}$ neocortex (Ctx) and hippocampus (Hpc) using LeafCutter (*Li et al., 2018*) (visualized with LeafViz **Knowles, 2022**) demonstrates reduction in exclusion of a 3' exon within the *Sort1* transcript. Diagram in (**A**) shows full-length *Sort1* gene in upper panel and highlights the differentially spliced region in gray. Chromosomal location, intron start and end points, annotation status, and Δ percent spliced in (dPSI) of the intronic region is listed in the lower table. qPCR analysis of additional WT and TDP-43$^{KQ/KQ}$ neocortex (**B**) and hippocampus (**C**) samples using primers specific for *Sort1* splice variants to show the ratio of *Sort1-ex17b:Sort1-WT* mRNA levels and total *Sort1* mRNA levels [**B** neocortex; *Sort1-ex17b:Sort1-WT* ratio, $F(1,22) = 51.85$, $p < 0.0001$; *Sort1* total, $F(1,25) = 0.01856$, $p = 0.8927$; **C** hippocampus; *Sort1-ex17b:Sort1-WT* ratio, $F(1,14) = 39.54$, $p < 0.0001$, *Sort1* total $F(1,16) = 2.086$, $p = 0.1679$]. (**D–G**) Images of western blots probed for Sort1 protein from neocortex (**D**) and hippocampus (**E**) lysates from 12- and 18-month-old mice. The Sort1 protein band intensity is plotted relative to total transferred protein (TTP) and quantified in **F, G** [neocortex, $F(1,14) = 9.308$, $p = 0.0086$; hippocampus, $F(1,13) = 6.117$, $p = 0.0280$]. (**A**) Neocortex $n = 5$ TDP-43$^{wt}$, $n = 6$ TDP-43$^{KQ/KQ}$; hippocampus $n = 4$ TDP-43$^{wt}$, $n = 5$ TDP-43$^{KQ/KQ}$. (**B**) Neocortex $n = 3$ 12-month TDP-43$^{wt}$, $n = 5$ 12-month TDP-43$^{KQ/KQ}$, $n = 9$ 18-month TDP-43$^{wt}$, $n = 9$ 18-month TDP-43$^{KQ/KQ}$. (**C**) Hippocampus $n = 3$ 12-month TDP-43$^{wt}$, $n = 5$ 12-month TDP-43$^{KQ/KQ}$, $n = 5$ 18-month TDP-43$^{wt}$, $n = 5$ 18-month TDP-43$^{KQ/KQ}$. (**D–G**) $n = 4$ 12-month TDP-43$^{wt}$, $n = 5$ 12-month TDP-43$^{KQ/KQ}$, $n = 4$ 18-month TDP-43$^{wt}$, $n = 4$ (hippocampus) or $n = 5$ (neocortex) 18-month TDP-43$^{KQ/KQ}$. Data are presented as mean ± standard deviation (SD). Two-way analysis of variance (ANOVA) followed by Šídák's multiple comparisons test. *F* statistics represent main effect of genotype, unless otherwise stated.

*Figure 8 continued on next page*

Figure 8 continued

Statistical significance is represented by asterisks, *$p < 0.05$, **$p < 0.01$, ***$p < 0.001$, ****$p < 0.0001$. Further statistical information is located in *Figure 8—source data 1* file. Figures showing uncropped western blot images are available in *Figure 8—source data 2* files. Full images of western blots are available in *Figure 8—source data 3* files.

The online version of this article includes the following source data for figure 8:

**Source data 1.** Complete statistical results and information for quantitative elements shown in *Figure 8*.

**Source data 2.** Figures showing uncropped western blot images of those presented in *Figure 8D,E*.

**Source data 3.** Full uncropped images of western blots shown in *Figure 8*.

explain why acetylation, or acetylation-mimic mutations within RRM1, alter TDP-43 dynamics and lead to pathology. Indeed, RNA-binding deficient TDP-43, including RRM mutations or deletions, showed increased TDP-43 phase separation, aggregation, nuclear egress, and neurotoxicity (*Kuo et al., 2014*; *Garcia Morato et al., 2022*; *Yu et al., 2021*; *Chiang et al., 2016*). Conversely, enhancing RNA binding was suggested to mitigate these phenotypes (*Mann and Donnelly, 2021*; *Mann et al., 2019*; *Grese et al., 2021*), in part due to TDP-43 retention within the nucleus (*Duan et al., 2022*). Thus, impaired RNA binding achieved by excessive lysine acetylation, or other RRM modifications, likely promotes TDP-43 pathogenesis in the form of mislocalization, aggregation, RNA dysregulation, and impaired autoregulation, as we observe with the TDP-43$^{K145Q}$ models described in this study.

Primary TDP-43$^{KQ/KQ}$ knock-in mouse and human iPSC-derived cortical neurons formed nuclear TDP-43+ foci that were exacerbated by acute oxidative stress. The acetylation-induced loss of RNA binding likely destabilized TDP-43 thus creating a more aggregate-prone conformation that can be further exacerbated by stressors (e.g, sodium arsenite) and likely other factors. Based on our original description of these TDP-43 foci, and several more recent studies of acetyl-mimic TDP-43 variants (*Garcia Morato et al., 2022*; *Yu et al., 2021*; *Keating et al., 2023*), the nuclear TDP-43+ foci observed in our neuronal models likely represent a spectrum of phase-separated structures, including anisosomes (*Yu et al., 2021*), and more immobile gel-like or solid aggregates. Future studies are needed to determine the molecular composition of these TDP-43+ foci, which could lead to a better understanding of their biogenesis, progression, and perhaps their dissolution. For example, it is conceivable that re-establishing TDP-43/RNA interactions could alleviate TDP-43 foci formation, suppress aggregation, and provide potential therapeutic avenues for us to consider (*Maharana et al., 2018*).

Homozygous TDP-43$^{KQ/KQ}$ knock-in mice showed impaired cognitive function, as well as behavioral disinhibition in the absence of overt motor deficits up to 18 months of age, supporting an FTLD-like phenotype in this model. The progressive behavioral defects and parallel increases in insoluble phosphorylated TDP-43 in the neocortex and hippocampus are also consistent with age as a driver of the FTLD-TDP phenotype. Even heterozygous TDP-43$^{wt/KQ}$ mice showed intermediate levels of both cognitive impairment and biochemical perturbations when compared to wild-type and homozygous littermates, suggesting a dose-dependent effect of TDP-43 acetylation. We were surprised to find no overt motor deficits at 18 months old despite the presence of mislocalized and phosphorylated TDP-43 in the spinal cord of TDP-43$^{KQ/KQ}$ mice. We speculate this could be due to any of the following non-mutually exclusive possibilities. (1) The levels of insoluble phosphorylated TDP-43 in the spinal cord have not crossed a critical threshold required for functional decline. (2) There is a time-dependent delay in the accumulation of TDP-43 and the onset of motor symptoms that could emerge beyond 18 months, which will require additional studies in aged cohorts of mice. (3) Spinal motor neurons in mice may possess some intrinsic resilience that suppresses functional decline. (4) It is possible that TDP-43$^{KQ/KQ}$ mice do indeed show subtle motor deficits that could be detected by more sensitive measures, such as electromyography-based techniques (*Arnold et al., 2015*; *Shefner, 2001*), for which future studies are warranted. (5) Finally, it remains plausible that phosphorylated TDP-43 is not inherently toxic or associated with functional decline in the spinal cord or other tissues (*Gruijs da Silva et al., 2022*; *Li et al., 2011*).

TDP-43$^{KQ/KQ}$ mice show striking similarities to other TDP-43 depletion models. For example, the RNA profiles in TDP-43$^{KQ/KQ}$ brain, particularly the hippocampus, were similar to a mouse model of striatal TDP-43 knock down (*Figure 7*), including upregulation of immune response genes (*Serpine1*, *Serping1*) and altered splicing of TDP-43 targets (*Kcnip2*, *Sort1*, *Dnajc5*, *Pdp1*, *Poldip3*) (*Polymenidou et al., 2011*). Similar splicing deficits were found in conditional CAMKIIa-driven *Tardbp* knockout

mice, accompanied by an FTLD-like phenotype with disinhibitory behavior and spatial learning deficits resembling those seen in TDP-43$^{KQ/KQ}$ mice (*Wu et al., 2019*; *LaClair et al., 2016*). Hippocampal knockdown of TDP-43 in adult mice impaired learning and memory, likely due to synaptic loss (*Ni et al., 2023*). This aligns with our finding of cognitive decline without overt cortical neuron loss or neurodegeneration. We suspect that TDP-43 loss of function causes synaptic dysfunction that may eventually lead to neurodegeneration beyond 18 months. In contrast to the lethality associated with complete TDP-43 depletion (*Kraemer et al., 2010*), the K145Q substitution creates a powerful yet viable partial loss of function model, which lacks developmental defects and ALS-like motor phenotypes. In addition, transcript profiles in TDP-43$^{KQ/KQ}$ mice do not entirely align with striatal TDP-43 knockdown (*Polymenidou et al., 2011*). This is likely in part because the K145Q mutation retains some level of RNA-binding capacity, leaving some TDP-43-dependent processes unperturbed. Acetylated TDP-43 could also lead to gain-of-function toxicity due to altered RNA-binding patterns and conformational instability, leading to nuclear export, enhanced phosphorylation and aggregation, and a dysregulated transcriptome, as we observe in TDP-43$^{KQ/KQ}$ mice. Both loss- and gain-of-function effects have been proposed for familial disease-causing *TARDBP* mutations (*Cascella et al., 2016*; *Diaper et al., 2013*; *Arnold et al., 2013*; *Halliday et al., 2012*), and TDP-43 acetylation may act in a similar manner.

RNA sequencing revealed profound transcriptomic alterations within aged TDP-43$^{KQ/KQ}$ mouse brains. Many up- and downregulated transcripts identified in acetylation-mimic mice were also differentially expressed in FTLD-TDP human tissue (*Figure 7*), particularly genes associated with cellular stress response, synaptic regulation, apoptotic signaling, and cellular adhesion (*Hasan et al., 2022*; *Gerrits et al., 2022*). Disease-related alternative splicing events were also common in TDP-43$^{KQ/KQ}$ neurons and mouse brain. For example, the altered splicing of *Sort1* and subsequent generation of *Sort1+ex17b* observed in TDP-43 acetylation-mimic mice (*Figure 8*) are also observed in FTLD-TDP brain (*Prudencio et al., 2012*; *Hu et al., 2010*). Moreover, SORT1 facilitates the endocytosis and lysosomal degradation of PGRN (*Hu et al., 2010*; *Pallesen and Vaegter, 2012*), a protein whose deficiency is causative of 10–30% familial FTLD cases (*Baker et al., 2006*; *Sieben et al., 2012*; *Greaves and Rohrer, 2019*). In humans, SORT1+ex17b contains an additional proteolytic site, resulting in the production of a soluble toxic variant that impairs the function of SORT1, prevents binding to PGRN and impacts neuronal survival (*Prudencio et al., 2012*; *Carlo et al., 2014*). A recent study showed that TDP-43 depletion increased SORT1+ex17b, impaired brain-derived neurotrophic factor (BDNF) signaling, and reduced synaptic plasticity in mice (*Tann et al., 2019*). Importantly, hiPSC-derived cortical neurons expressing acetylation-mimic TDP-43$^{K145Q}$ showed impaired splicing of putative FTLD-ALS spectrum disease biomarkers *UNC13A* and *STMN2* (*Brown et al., 2022*; *Prudencio et al., 2020*; *Melamed et al., 2019*; *Ma et al., 2022*). We speculate that acetylated TDP-43 alters SORT1/PGRN signaling, as well as other pathways critical to neurotransmission, contributing to the synaptic dysregulation that is so evident in our transcriptome data. Future mechanistic studies are aimed at interrogating SORT1-PGRN, UNC13A, and STMN2-related signaling pathways in TDP-43$^{KQ/KQ}$ mice and hiPSC-derived neurons.

Other aberrantly spliced transcripts in TDP-43$^{KQ/KQ}$ mice provide additional insight into pathogenic mechanisms. The synaptic regulators *Arfgef2* (*Sheen et al., 2004*) and *Arhgap10* (*Sekiguchi et al., 2020*; *Cuttler et al., 2021*) were among the most affected transcripts. Concurrently, *Sema5b* which regulates synaptic connections and axon guidance (*Duan et al., 2014*; *Jung et al., 2019*; *O'Connor et al., 2009*; *Kantor et al., 2004*), was strongly downregulated in TDP-43$^{KQ/KQ}$ hippocampus and cortex, as were other genes involved in synapse homeostasis and signaling. Furthermore, splicing of LRP8 (low-density lipoprotein receptor-related protein 8) was significantly altered in TDP-43$^{KQ/KQ}$ mice, and abnormalities in LRP8 can cause learning and memory defects, likely by disrupting Reelin-mediated synaptic plasticity and long-term potentiation (LTP) (*Telese et al., 2015*; *Beffert et al., 2005*; *Reddy et al., 2011*; *Hinrich et al., 2016*). Similarly, *Nos1*, encoding the neuronal nitric oxide synthase (nNOS) protein that regulates LTP and synaptic plasticity via the production of nitric oxide (*Hardingham et al., 2013*; *Calabrese et al., 2007*), showed aberrant splicing patterns in TDP-43$^{KQ/KQ}$ mice. Importantly, all of these synaptic regulators (*Nos1, Lrp8, Argef2, Sema5b*) and other significantly altered splice variants (e.g., *Adipor2, Mapk14, Smarca4, Sort1, Mapt*) have been linked to AD and other neurodegenerative diseases (*Park et al., 2016*; *Hinrich et al., 2016*; *Calabrese et al., 2007*; *Sun et al., 2020*; *Zhu et al., 2001*; *Alam and Scheper, 2016*; *Waragai et al., 2017*; *Shi et al., 2021*;

Guix et al., 2005; Chong et al., 2018; Seto et al., 2022). Our transcriptome data strongly suggest that splicing and transcriptional abnormalities due to TDP-43 acetylation impacts synaptic plasticity, neurotransmission, and neuronal survival pathways in the TDP-43$^{KQ/KQ}$ model, which overlaps with abnormalities found in human FTLD-TDP patients. We suspect that this prominent RNA signature may underlie the cognitive decline in TDP-43$^{KQ/KQ}$ neocortex and hippocampus.

Given the correlation between TDP-43 acetylation and FTLD-TDP presented in this study, it is perhaps surprising that acetylated TDP-43 (detected with an acetylation-specific ac-K145 antibody) is found within ALS spinal cord inclusions but has not yet been detected in FTLD-TDP cortex (Cohen et al., 2015). However, TDP-43 pathology in FTLD-TDP cases is dominated by the presence of C-terminal fragments lacking RRM1 thereby precluding detection of the acetylated K145 site (Igaz et al., 2008; Berning and Walker, 2019). It remains possible that TDP-43 acetylation still occurs in FTLD-TDP at one or more additional lysine residues, and thus future mass spectrometry studies will be needed to determine whether a TDP-43 acetylation profile occurs earlier in the progression of FTLD-TDP prior to C-terminal fragmentation.

In conclusion, disrupting RNA binding with an acetylation-mimic TDP-43$^{K145Q}$ mutation results in an age-dependent, dementia-like phenotype characterized by signatures of FTLD, including progressive cognitive deficits, TDP-43 mislocalization and phosphorylation, impaired autoregulation, and prominent RNA dysregulation. TDP-43 acetylation affected many known FTLD-associated pathways, including STMN2, SORT1, and stress response signaling, while also pinpointing new putative pathways as mediators of disease pathogenesis. By developing models of TDP-43 acetylation, we expand our understanding of sporadic TDP-43 proteinopathies and provide new avenues to identify therapeutics that target the pathogenic mechanisms underlying TDP-43 proteinopathies.

## Methods

### Animal husbandry

Mice were housed in ventilated microbarrier cages on racks providing high efficiency particulate air (HEPA)-filtered air supply to each cage. Animals were kept on a 12-hr light–dark cycle with access to food and water ad libitum. All animal husbandry, experiments, and procedures were performed in strict compliance with animal protocols approved by the Institutional Animal Care and Use Committee (IACUC) of the University of North Carolina at Chapel Hill (Protocol #21.257).

### Primary neuron cultures

Murine primary cortical neurons were performed using wild-type C57Bl/6 mice (Charles River) or TDP-43$^{wt/KQ}$ breeding pairs. Timed pregnant females at embryonic days 15–16 were lethally anesthetized with isoflurane. The abdominal cavity was opened, and the uterus incised to remove the placentas and embryonic sacs, which were washed briefly in ice-cold 70% ethanol and then placed in cold 4-(2-hydroxyethyl)-1-piperazineethanesulfonic acid (HEPES)-buffered Bank's balanced salt solution (HBSS). The embryos were transferred into a 10-cm dish containing cold HBSS, the embryos removed from the amniotic sac, and the brains were extracted from the cranium. For TDP-43$^{wt/KQ}$ dissections, process was paused for approximately 2 hr to permit genotyping of the fetuses, during which the brains were stored at 4°C protected from light in a Hibernate-E (BrainBits NC9063748) solution supplemented with B27 (Gibco 17504044) and GlutaMAX (Thermo Fisher 35050061). After genotyping, if applicable, the cerebral cortices from each brain were isolated under a stereomicroscope, pooled by genotype, minced with forceps, and digested for 30 min at 37°C in a filter-sterilized HBSS solution containing 20 U/ml papain (Worthington Biochemical LS003126), 1 mM EDTA, 0.2 mg/ml L-cysteine, and 5 U/ml DNAse (Promega M6101). The enzyme solution was removed, and the digested tissue was washed twice with sterile HBSS. Warm plating media [BrainPhys media (Stemcell 05790), 5% fetal bovine serum, 1× penicillin/streptomycin (Gibco 15140122), 1× B27, 1× GlutaMAX] containing 5 U/ml DNAse, was added and the tissue was dissociated mechanically using a P1000 pipette. The resulting cell suspension was spun down for 5 min at 1.5 rcf to pellet the cells, resuspended in plating media, and filtered through a 40-mm cell strainer. Cells were counted using a hemocytometer and plated onto poly-D-lysine (PDL)-coated 12-well tissue culture plates (Corning 356470) at 300 K cells/well (for RNA extraction) or 96-well glass bottom black wall plates (Cellvis P96-1.5H-N) at 30 K/well (for immunofluorescence and microscopy). 16–24 hr after plating, all plating media was removed and replaced

with neuronal cell media (BrainPhys, 1× GlutaMAX, 1× B27, 1× penicillin/streptomycin). Cultures were incubated at 37°C, 5% $CO_2$ and 95% humidity with half-media exchanges every 3 days for the duration of all experiments.

## Lentivirus preparation and neuron transduction

Lentiviral vectors were constructed in-house using the pUltra vector as a backbone for cloning via restriction enzyme digestion followed by ligation. The pUltra construct was acquired from Addgene (gift from Malcolm Moore; Addgene plasmid # 24129; http://n2t.net/addgene:24129; RRID:Addgene_24129) (*Lou et al., 2012*). To generate TDP-43 constructs, TDP-43 variant gene fragments were PCR amplified using TDP-43 F' and R' primers (*Supplementary file 3*) and a pcDNA5/TO- myc-TDP-43[wt], -myc-TDP-43[K145Q], or -myc-TDP-43[K145R] plasmid (*Cohen et al., 2015*) as the template. The pUltra construct was digested with AgeI and BamHI, and then incubated with the appropriate TDP-43 PCR product in the presence of T4 DNA ligase and T4 Polynucleotide Kinase for 1 hr at room temperature (RT). The resulting ligation product was transformed into NEB stable competent cells using standard protocols.

Lentiviral production was performed by co-transfecting (CalPhos Mammalian Transfection Kit, TakaRa 631312) 37.5 µg lenti-plasmid with 25 µg psPAX2, 12.5 µg VSVG, and 6.25 µg REV for each 15 cm dish of lenti-X 293T cells (Takara 632180) and 3 dishes of cells were used for each lentiviral production. Three days after transfection, culture media were collected and centrifuged at 2000 × *g* for 10 min. Lentiviral particles were purified using a double-sucrose gradient method. Briefly, the supernatants were loaded onto a 70–60−30–20% sucrose gradient and centrifuged at 70,000 × *g* for 2 hr at 17°C (Beckman Optima LE-80K Ultracentrifuge, SW 32 Ti Swinging-Bucket Rotor). The 30–60% fraction containing the viral particles was retrieved, resuspended in phosphate-buffered saline (PBS), filtered with a 0.45-µm filter flask before loaded onto a 20% sucrose cushion, and centrifuged a second time at 70,000 × *g* for 2 hr at 17°C. The supernatants were carefully discarded, and the viral particles present in the pellet were resuspended in PBS, aliquoted and stored at −80°C.

Lentivirus aliquots were evaluated for neuronal transduction ability prior to each experiment and diluted to approximately equal concentrations of effective virus. Lentiviral transductions were performed on DIV10 for a DIV14 harvest or DIV24 for a DIV28 harvest. In brief, lentiviruses were diluted to a 2× concentration in neuronal media, and then added to plates via half-media exchange. Twenty-four hours after viral transduction, all virus-containing media was removed and replaced with 50% fresh neuronal media and 50% conditioned media from plates containing WT non-transduced neurons. Neurons were then cultured to DIV14 or DIV28 as the experiment required.

## hiPSC-derived cortical neurons
### iPSC maintenance
The HPC26 iPSCs were maintained on Matrigel-coated dishes (Corning 354480) in StemFlex Medium (Thermo Fisher Scientific A3349401) and passaged every 3–4 days with 0.5 mM EDTA dissociation solution as previously described (*Beltran et al., 2021*).

### CRISPR/Cas9 genome editing
We used a control wild-type HPC26 cell line (*Beltran et al., 2021*) to generate two separate point mutations in the TARDBP gene that encodes the TDP-43 protein. Both mutations created single amino acid substitutions to change lysine 145 to glycine or arginine (K145Q and K145R). Benchling was used to design two guide RNAs and the corresponding single-stranded DNA donor oligos. The guide RNAs were purchased from Synthego, and the donor oligos from IDT. Genome editing was performed as described in *Battaglia et al., 2019*. Briefly, $3 \times 10^5$ iPSCs were electroporated on the Neon electroporation system (Thermo Fisher Scientific) with recombinant protein complexes made of Cas9 v2 protein (Thermo Fisher Scientific A36498), 900 ng sgRNA-TARDBP and 2700 ng of single-stranded donor oligonucleotide-K145Q or donor oligonucleotide-K145R (*Supplementary file 3*). Seventy-two hours after electroporation, cells were plated for single-cell screening on a 96-well plate format using the limited dilution method. After 2 weeks, single cells were expanded, genome DNA collected and the exon 4 of TARDBP amplified using the specific primers Ex4Fr and Ex4Rv (*Supplementary file 3*). Screening for single or double allele gene edits was performed by Sanger sequencing (*Figure 3— figure supplement 1a–c*).

## Characterization of edited iPSC clones

Stemness edited iPSCs were assessed by immunofluorescent staining of the pluripotency factors OCT4, SOX2, SSEA4, and Tra-1–60 using specific antibodies (*Supplementary file 4*) as described (*Beltran et al., 2021*; *Figure 3—figure supplement 1d, e*). To confirmed stemness and differentiation capabilities of the edited iPSCs we used the Taqman hPSC Scorecard (Thermo Fisher A15871) as described in *Battaglia et al., 2019*. Briefly, iPSCs were differentiated into all three germ layers using STEMdiff Trilineage Differentiation Kit (StemCell Technologies), a monolayer-based protocol to directly differentiate hES cells in parallel into the three germ layers. Non-differentiated and differentiated cells were lysed and total RNA purified using the RNeasy kit (QIAGEN 74004). RNA reversed transcription was performed with the high-capacity cDNA Reverse Transcription kit (Thermo Fisher 4368813) following the Taqman Scorecard's manufacture guidelines. qRT-PCR was carried out using the QuantStudio 7 Flex Real-Time PCR system. The TaqMan PCR assay combines DNA methylation mapping, gene expression profiling, and transcript counting of lineage marker genes (*Bock et al., 2011*).

## iPSC-derived cortical neurons

We adapted, modified, and standardized a protocol to generate mature cortical neurons from iPSCs using a dual SMAD inhibition protocol (*Chambers et al., 2009*; *Shi et al., 2012*). First, undifferentiated iPSCs were collected with Accutase, counted and $3 \times 10^5$ cells cultured in StemScale PSC medium (A4965001) supplemented with 10 µM Y27632 (PeproTech 1293823) and cultured in suspension on an orbital shaker at 37°C and 5% $CO_2$. After 48 hr, cells were dissociated with Accutase and $3 \times 10^5$ cells differentiated into NPCs as neutrosphere in StemScale PSC medium supplemented with 10 µM Y27632 (PeproTech 1293823) and 1.5 µM CHIR99021 (PeproTech 2520691), 10 µM SB431542 (PeproTech 3014193) and 50 nM LDN-193189 (Sigma SML0755). CHIR99021 was removed after 24 hr and cells were cultured for 10 days with daily medium changes and neurospheres dissociated at 1:3 ratio twice a week. Then NPCs were expanded for 7 days in the presence of 20 ng/ml FGF (Peprotech 100-18B) in Neuronal Expansion Medium [1:1 Advanced Dulbecco's Modified Eagle Medium (DMEM)/ F12 (12634028) and Neural Induction Supplement (A1647801)]. NPCs were plated for maturation on PDL/Laminin-coated plates or coverslips in cortical neuron maturation medium [1:1 Advanced DMEM/ F12 and Neurobasal; 1× GlutaMAX, 100 mM B-mercaptoethanol, 1× B27, 0.5× N2, 1× non-essential amino acids (NEAA), and 2.5 mg/ml insulin (Sigma-Aldrich I9278)] supplemented with 10 ng/ml BDNF, 10 ng/ml glial cell-derived neurotrophic factor (GDNF), and 10 µM N-[N-(3,5-difluorophenacetyl)-L-alanyl]-S-phenylglycine t-butyl ester (DAPT). Seventy-two hours after plating, cells were treated with 1 µg/ml Mitomycin C for (Sigma M5353) for 1 hr. Cellular identity was assessed specific markers of neuronal progenitor (PAX6, SOX2, and SOX1), and mature neurons (MAP2, TUJ1, CTIP2, and SATB2) (*Figure 3—figure supplement 2*; *Supplementary file 4*). All reagents were purchased from Thermo Fisher unless otherwise noted.

## Primary mouse neuron and hiPSC-derived cortical neuron arsenite treatments and immunocytochemistry

On DIV14 (mouse primary neurons) or mature hiPSC-derived cortical neurons (~DIV50), neurons were treated for 2 hr with 200 µM sodium arsenite ($NaAsO_2$) or vehicle (molecular biology grade water). Following treatment, neurons were fixed with 4% paraformaldehyde (PFA) in 1× PBS and washed with 1× PBS. Neurons were permeabilized with 0.3% Triton X-100 in 1× PBS for 15 min at RT, blocked in an 8% normal goat serum (NGS) 0.2% Triton X-100 in 1xPBS solution for 1–2 hr at RT, and incubated with primary antibodies (*Supplementary file 4*) diluted in 4% NGS, 0.2% Triton X-100, 1×xPBS solution overnight at 4°C. The next day, the neurons were washed with 1xPBS, followed by application of fluorescent secondary antibodies (*Supplementary file 4*) diluted in 4% NGS, 0.2% Triton X-100, 1xPBS solution for 2 hr at RT, covered. Cells were washed four times with 1× PBS, with the third wash containing 1 µg/ml 4',6-diamidine-2'-phenylindole dihydrochloride (DAPI). 96-Well plates were preserved in an 85% glycerol in 1× PBS solution containing 0.4% sodium azide for microbial prevention. Coverslips were mounted onto pre-cleaned glass slides using ProLong Diamond Antifade Mountant (Invitrogen P36961).

## Cultured neuron fluorescence microscopy and image analysis

Mouse primary cortical neurons in 96-well plates were visualized via automated fluorescence microscopy using EVOS M7000 Imaging System (Thermo Fisher AMF7000) equipped with Olympus ×20/0.75 NA UPlanSApo and Olympus ×40/0.95 NA UPlanSApo objectives, and the following filter cubes: DAPI (357/44 nm Excitation; 447/60 nm Emission), GFP (482/25 nm Excitation; 524/24 nm Emission), Texas Red (585/29 nm Excitation; 628/32 nm Emission), and Cy5.5 (655/40 nm Excitation; 794/160 nm Emission). Images were acquired in an automated fashion, using DAPI fluorescence as the autofocus substrate, taking 16–32 images per well of 2–3 wells per experimental condition. Automated quantitative image analysis was performed using CellProfiler 4.0 (*Carpenter et al., 2006*; *Stirling et al., 2021*) to measure TDP-43 fluorescence intensity within neuronal compartments and to count TDP-43-positive foci within neurons. To identify neurons and subcellular compartments, DAPI was used to delineate nuclei and NeuN was used to label neuronal soma. hiPSC-derived cortical neurons were imaged on a Leica SP8X Falcon confocal microscope equipped with a ×63/1.40 NA Plan Apochromatic (oil) objective and hybrid GaAsP detectors, Leica Application Suite X Life Sciences software (Leica, Wetzlar, Germany).

## Mouse model generation and genotyping

### CRISPR/Cas9 reagents

Cas9 guide RNAs targeting the mouse *Tardbp* K145 codon were identified using Benchling software (Benchling, San Francisco, CA, USA). Three guide RNAs were selected for activity testing. Guide RNAs were cloned into a T7 promoter vector followed by in vitro transcription and spin column purification (RNeasy, QIAGEN). Guide RNAs were tested for cleavage activity by in vitro cleavage assay. Each guide RNA was incubated with Cas9 protein (UNC Protein Expression and Purification Core Facility) and PCR-amplified guide RNA target site. The products were run on an agarose gel for analysis of target site cleavage. Based on this assay, the guide RNA selected for genome editing in embryos was Tardbp-g79T (*Supplementary file 3*). The donor oligonucleotide for insertion of the K145Q and silent genotyping mutations was Tardbp-K145Q-T (*Supplementary file 3*). Two silent mutations were induced to create a unique HinfI restriction enzyme digestion site and permit genotyping by PCR and gel electrophoresis.

C57BL/6J zygotes were microinjected with microinjection buffer (5 mM Tris–HCl pH 7.5, 0.1 mM EDTA) containing (1) 20 ng/µl Cas9 mRNA, 10 ng/µl g79T guide RNA, and 50 ng/µl donor oligonucleotide (founder #9) or (2) 20 ng/µl Cas9 mRNA, 400 nM Cas9 protein, 10 ng/µl g79T guide RNA, and 50 ng/µl donor oligonucleotide (founders #23, 26, 39, 44, 53). Injected embryos were implanted in recipient pseudopregnant females and resulting pups were screened by PCR and sequencing for the presence of the desired mutant allele. Founder lines were propagated as heterozygotes and regularly sequenced to confirm retention of the K145Q and silent mutations.

We assessed the likelihood of off-target mutations using prediction algorithms to ensure low likelihood of off-target effects. The Benchling MIT off-target score for Tardbp-g79T is 72 (*Anderson et al., 2018*), and the CRISPOR-generated MIT off-target score is 86. CRISPOR also gave a Cutting frequency determination score of 86 (*Concordet and Haeussler, 2018*; *Haeussler et al., 2016*; *Doench et al., 2016*). Moreover, all of the predicted exonic off-target sites from CRISPOR have four mismatches, making them unlikely to be mutated. Sequencing of the founder lines and their progeny did not detect any insertions or deletions within a 550-bp region surrounding the *Tardbp* locus.

### TDP-43$^{K145Q}$ mouse line genotyping

Each mouse was genotyped prior to any experimental use. DNA was extracted from ear punch or toe clip tissue using the HotSHOT method (*Truett et al., 2000*). PCR amplification of the modified *Tardbp* locus was performed with ApexRed master mix (Genesee Scientific 42-138B) and *TDPKQ* F′ and *TDPKQ* R′ primers (*Supplementary file 3*) using the following cycling parameters: 95°C denaturation for 3 min; 15 cycles of 95°C for 30 s, 72°C for 30 s then −1°C per cycle, 72°C 60 s; 25 cycles of 95°C for 30 s, 58°C for 30 s, 72°C for 60 s; 72°C extension for 5 min. The PCR product was then incubated with HinfI enzyme (New England Biosystems R0155S) in CutSmart buffer (New England Biosystems B7204) for 1 hr at 37°C. Digested PCR products were separated on a 2% agarose Tris–acetate–EDTA gel containing SybrSafe stain (Invitrogen S33102) and visualized on an ImageQuant LAS4000 machine.

### Mouse behavior

All testing was performed by experimenters blinded to mouse genotype.

### Open field

Exploratory activity in a novel environment was assessed by a 1-hr trial in an open-field chamber (41 cm × 41 cm × 30 cm) crossed by a grid of photobeams (VersaMax system, AccuScan Instruments). Counts were taken of the number of photobeams broken during the trial in 5-min intervals, with separate measures for locomotor activity (total distance traveled) and vertical rearing movements. Time spent in the center region was used as an index of anxiety-like behavior.

### Conditioned fear

Mice were evaluated for conditioned fear using the near-infrared image tracking system (MED Associates, Burlington, VT). The procedure had the following phases: training on day 1, a test for context-dependent learning on day 2, and a test for cue-dependent learning on day 3. *Training.* On day 1, mice were placed in the test chambers, contained in sound-attenuating boxes. The mice were allowed to explore the novel chambers for 2 min before presentation of a 30-s tone (80 dB), which co-terminated with a 2-s scrambled foot shock (0.4 mA). Mice received two additional shock-tone pairings, with 80 s between each pairing, and were removed from the test chambers 80 s following the third shock. *Context- and cue-dependent learning.* On day 2, mice were placed back into the original conditioning chambers for a test of contextual learning. Levels of freezing (immobility) were determined across a 5-min session. On day 3, mice were evaluated for associative learning to the auditory cue in another 5 min session. The conditioning chambers were modified using a Plexiglas insert to change the wall and floor surface, and a novel odor (dilute vanilla flavoring) was added to the sound-attenuating box. Mice were placed in the modified chamber and allowed to explore. After 2 min, the acoustic stimulus (an 80-dB tone) was presented for a 3-min period. Levels of freezing before and during the stimulus were obtained by the image tracking system.

### Grip strength

Grip strength was evaluated using precision force gauges. Measures were based on paw grasp of a metal grid by a mouse gently pulled by the tail. Digital force meters (Chatillon DFIS-10; Largo, FL) were mounted on an acrylic platform (San Diego Instruments), with two different grids: a left-hand grid for the front paws, and a right-hand grid for all-four paws. Each grid connected to a force transducer, which provided measures of peak force (newtons). Each mouse was given three trials, with at least 1 min between each trial. Each trial had two components: front-paw measures from the left-hand grid, immediately followed by all-four-paw measures from the right-hand grid.

### Rotarod

Subjects were tested for motor coordination and motor learning on an accelerating rotarod (Ugo Basile, Stoelting Co, Wood Dale, IL). For the first test, mice were given three trials, with 45 s between each trial. Two additional trials were given 48 hr later. Rpm (revolutions per minute) was set at an initial value of 3, with a progressive increase to a maximum of 30 rpm across 5 min (the maximum trial length). Measures were taken for latency to fall from the top of the rotating barrel.

### Morris water maze

The water maze was used to assess spatial and reversal learning, swimming ability, and vision. The water maze consisted of a large circular pool (diameter = 122 cm) partially filled with water (45 cm deep, 24–26°C), located in a room with numerous visual cues. The procedure involved three separate phases: a visible platform test, acquisition in the hidden platform task, and a test for reversal learning (an index of cognitive flexibility).

### Visible platform test

Each mouse was given 4 trials per day, across 2 days, to swim to an escape platform cued by a patterned cylinder extending above the surface of the water. For each trial, the mouse was placed in the pool at one of four possible locations (randomly ordered), and then given 60 s to find the visible

platform. If the mouse found the platform, the trial ended, and the animal was allowed to remain 10 s on the platform before the next trial began. If the platform was not found, the mouse was placed on the platform for 10 s, and then given the next trial. Measures were taken of latency to find the platform and swimming speed via an automated tracking system (Noldus Ethovision).

## Acquisition and reversal learning in a hidden platform task

Following the visible platform task, mice were tested for their ability to find a submerged, hidden escape platform (diameter = 12 cm). Each mouse was given 4 trials per day, with 1 min per trial, to swim to the hidden platform. The criterion for learning was an average group latency of 15 s or less to locate the platform. Mice were tested until the group reached criterion, with a maximum of 9 days of testing. When the group reached criterion (on day 5 in the present study), mice were given a 1-min probe trial in the pool with the platform removed. Selective quadrant search was evaluated by measuring the number of crosses over the location where the platform (the target) had been placed during training, versus the corresponding area in the opposite quadrant. Following the acquisition phase, mice were tested for reversal learning, using the same procedure as described above. In this phase, the hidden platform was re-located to the opposite quadrant in the pool. As before, measures were taken of latency to find the platform. On day 5 of testing, the platform was removed from the pool, and the group was given a probe trial to evaluate reversal learning.

## Acoustic startle test

This procedure was used to assess auditory function, reactivity to environmental stimuli, and sensorimotor gating. The test was based on the reflexive whole-body flinch, or startle response, that follows exposure to a sudden noise. Measures were taken of startle magnitude and PPI, which occurs when a weak prestimulus leads to a reduced startle in response to a subsequent louder noise.

Mice were placed into individual small Plexiglas cylinders within larger, sound-attenuating chambers. Each cylinder was seated upon a piezoelectric transducer, which allowed vibrations to be quantified and displayed on a computer (San Diego Instruments SR-Lab system). The chambers included a ceiling light, fan, and a loudspeaker for the acoustic stimuli. Background sound levels (70 dB) and calibration of the acoustic stimuli were confirmed with a digital sound level meter (San Diego Instruments). Each session began with a 5-min habituation period, followed by 42 trials of 7 different types: no-stimulus (NoS) trials, trials with the acoustic startle stimulus (AS; 40 ms, 120 dB) alone, and trials in which a prepulse stimulus (20 ms; either 74, 78, 82, 86, or 90 dB) occurred 100 ms before the onset of the startle stimulus. Measures were taken of the startle amplitude for each trial across a 65 ms sampling window, and an overall analysis was performed for each subject's data for levels of PPI at each prepulse sound level (calculated as $100 - [(\text{response amplitude for prepulse stimulus and startle stimulus together/response amplitude for startle stimulus alone}) \times 100]$).

## Tissue harvest and preparation

At each end point, mice were anesthetized deeply with isoflurane and euthanized via rapid decapitation. The hippocampus and neocortex were dissected out of the brain on a cold surface using clean surgical tools, placed into cryo-safe nuclease-free microcentrifuge tubes, flash frozen in liquid nitrogen (LN2), and stored at −80°C until processing. Frozen tissue was then pulverized in LN2-cooled stainless steel Cryo-Cups using cold stainless-steel pestles and transferred into cold nuclease-free microcentrifuge tubes and immediately stored at −80°C until use.

## Solubility fractionation, nucleo-cytoplasmic fractionation, and immunoblotting

All steps of protein fractionation were performed on ice unless otherwise indicated. For solubility fractionation, pulverized tissue was suspended in 5 µl/mg of ice-cold 1× RIPA buffer (50 mM Tris pH 8.0, 150 mM NaCl, 1% NP-40, 5 mM EDTA, 0.5% sodium deoxycholate, 0.1% sodium dodecyl sulfate [SDS]) containing a mix of protease, phosphatase, and deacetylase inhibitors [1 µg/ml Peptstatin A (Sigma P4265), 1 µg/ml Leupeptin (Sigma L2023), 1 µg/ml $N_\alpha$-Tosyl-L-lysine chloromethyl ketone hydrochloride (TPCK) (Sigma T7254), 1 µg/ml Trypsin inhibitor (Sigma T9003), 1 µg/mL N-p-Tosyl-L-phenylalanine chloromethyl ketone (TLCK) (Sigma T4376), 0.67 µg/ml trichostatin A, 10 mM nicotidamide, 1 mM phenyline thanosulfyl fluoride, 1 mM phenylmethylsulfonyl fluoride]. The solution

was homogenized by sonication and centrifuged at 4°C for 45 min at 18,000 × rcf. The supernatant was removed and saved as the RIPA-soluble (soluble) protein fraction. The pellet was resuspended in RIPA buffer with inhibitor mixture, sonicated, and centrifuged as described above, and the supernatant discarded. The resulting pellet of RIPA-insoluble material was resuspended in approximately 1 μl/μg of Urea buffer (7 M urea, 2 M thiourea, 4% CHAPS, 30 mM Tris, pH 8.5) with the inhibitor mixture (as above), sonicated to homogenize, and centrifuged at RT for 45 min at 21,000 × rcf. The resulting supernatant of RIPA-insoluble, urea-soluble protein fraction was saved as the 'insoluble' protein fraction.

Nuclear and cytoplasmic soluble proteins were isolated using the Thermo Fisher NE-PER Nuclear and Cytoplasmic Extraction Kit (Thermo Scientific 78835) per the manufacturer's instructions, with the addition of the protease, phosphatase, and deacetylase inhibitors (as above) to each buffer.

All soluble protein fractions were analyzed by Bicinchoninic acid (BCA) assay (Thermo Scientific 23225) to determine protein concentration. Equal quantities of protein per sample were run onto 4–20% Tris-Glycine SDS–polyacrylamide gel electrophoresis gels (Bio-Rad 5671095) under reducing conditions and then transferred onto nitrocellulose membranes. Total transferred protein (TTP) was assessed using PonceauS (Research Products International Corp P56200) protein stain on nitrocellulose membranes per the manufacturer's instructions. Membranes were washed three times in Tris-buffered saline with 0.1% Tween-20 (TBST), once in 1x Tris-buffered saline (TBS), and blocked in 2% nonfat milk in 1× TBS for 1–2 hr at RT. Membranes were incubated with primary antibodies (*Supplementary file 4*) diluted in 2% milk overnight at 4°C. The primary antibody solution was removed and membranes washed three times in TBST, once in TBS, and incubated with cross adsorbed horseradish peroxidase (HRP)-conjugated goat secondary antibodies (*Supplementary file 4*) diluted in 2% milk at 1:10,000 or 1:2000 for soluble or insoluble protein immunoblots, respectively, for 1–2 hr at RT. Blots were then visualized by chemiluminescent imaging using an ImageQuant LAS4000 machine. Densitometry analysis to quantify western blot images was performed in LI-COR Image Studio Lite (Lincoln, NE, USA).

## Mouse brain RNA isolation, RNA sequencing, and data analysis

Mouse brain tissue was isolated, flash frozen, and pulverized as described above. Approximately 20 mg of pulverized brain tissue per sample was used to isolate RNA. 1 ml of TRIzol (Invitrogen 15596018) was added to each nuclease-free Eppendorf tube containing pulverized tissue, and tissues were lysed via trituration with a P1000 pipette tip, followed by trituration with a 21-G and then a 25-G needle on a 1-ml syringe. Samples were centrifuged at 4°C for 5 min at 10,000 rcf to remove tissue debris. The supernatant was removed and added to a new tube containing 200 μL of chloroform, which were mixed by inversion and cooled on ice for 5 min. Samples were centrifuged at 4°C for 15 min at 10,000 rcf, and the upper aqueous phase containing RNA was transferred into a new tube, followed by the addition of 100 μl of isopropanol to precipitate RNA and overnight incubation at −20°C. The next day, samples were centrifuged at top speed (18,000 rcf) for 20 min at 4°C to pellet the RNA. The pellets were washed twice with 1 ml of ice-cold 70% molecular biology grade ethanol and then air dried for 15 min at RT. The RNA pellets were resuspended in 20 μl of nuclease-free water. On-column DNAse digestion and RNA clean-up were then performed using the QIAGEN RNeasy mini kit (QIAGEN, Inc 74106) per the manufacturer's instructions, followed by elution in nuclease-free water.

RNA concentration was assessed using Qubit RNA BR Assay Kit (Q10210) and a Qubit 3.0 Fluorometer. RNA integrity was assessed using an Agilent 4150 TapeStation system and associated RNA screen tape reagents (Agilent 5067-5576). Only samples with an estimated RNA integrity number (RIN) ≥7.0 were sent to the New York Genome Center (NYGC) for bulk total RNA sequencing. Upon receipt at NYGC, RNA samples were re-evaluated for quantification and integrity, using Ribogreen and Fragment Analyzer 5300, respectively. Total RNA libraries were prepped using Kapa Total library prep with Ribo-Erase, in accordance with manufacturer recommendations. Briefly, 500 ng of total RNA was used for ribosomal depletion and fragmentation of total RNA. Depleted RNA underwent first- and second-strand cDNA synthesis. cDNA was then adenylated, ligated to Illumina sequencing adapters, and amplified by PCR (using nine cycles). The cDNA libraries were quantified using Fragment Analyzer 5300 (Advanced Analytical) kit FA-NGS-HS (Agilent DNF-474-1000) and Spectramax M2 (Molecular Devices) kit Picogreen (Life Technologies P7589). Libraries were sequenced on an

Illumina NovaSeq sequencer, using paired end sequencing (2 × 100 bp cycles) to a depth of >75 M read pairs per sample.

Raw reads were then trimmed and filtered of adapter sequencing using cutadapt (*Martin, 2011*) and filtered such that at least 90% of bases had a quality score of at least 20. Reads were then aligned to the reference mouse genome (mm10, RefSeq gene annotations) using STAR v2.5.2b (*Dobin et al., 2013*), and transcript abundance was estimated using salmon (*Patro et al., 2017*). Differential expression between TDP43-KQ and TDP43-WT cortex and hippocampus was then detected using DESeq2 v1.34.0 (*Love et al., 2014*) in R v4.1.0 (*R Development Core Team, 2022*), using a design that corrects for both mouse sex and litter effects. These batch effects were also removed from the VST-normalized expression values using limma (*Ritchie et al., 2015*). The correlation between log2 fold change values between DEGs in the cortex and hippocampus was determined and Pearson correlation coefficient and two-sided p-value were computed by ggpubr stat_cor (*Kassambara, 2020*; *Wickham, 2016*), and the smoothed linear model was fit using geom_smooth with method="lm".

DEGs ($p_{adj}$ < 0.05) were then separated into 6 groups based on their intersections between the two brain regions using UpSetR v1.4.0 (*Conway et al., 2017*) and plotted with ComplexUpset (*Krassowski et al., 2021*). Normalized expression values were then centered around the mean of TDP43-WT for each respective brain region and plotted with ComplexHeatmap (*Gu et al., 2016*). GO enrichments were then assessed using gprofiler2 v0.2.1 (*Raudvere et al., 2019*; *Kolberg et al., 2020*) and summarized using simplifyEnrichment v1.7.2 (*Gu and Hübschmann, 2023*). Previously published DEGs from *Hasan et al., 2022* and *Polymenidou et al., 2011* were retrieved from the respective publications; significant over-enrichments as well as human gene symbol mappings to mouse orthologs were performed using gprofiler2 (g:Orth). Differential splicing analyses were performed on splice junctions extracted from genome-aligned BAM files using regtools and LeafCutter (*Li et al., 2018*), where tests compared TDP43-KQ to TDP43-WT correcting for sex for each brain region. Results were then summarized and visualized using LeafViz (*Knowles, 2022*).

## Quantitative Reverse Transcription-Polymerase Chain Reaction (RT-PCR)

RNA was isolated from mouse brain tissue as described above. For quantitative RT-PCR (RT-qPCR) experiments involving mouse primary cortical neurons or human iPSC-derived cortical neurons, cells were washed with 1× PBS, harvested from the culture dish, and RNA was extracted using the QIAGEN RNeasy mini kit (QIAGEN, Inc 74106) with on-column DNAse digestion per the manufacturer's instructions. RNA concentration was determined using a NanoDrop 2.0 spectrophotometer, 250 ng to 1 µg of RNA was used to generate cDNA using Applied Biosystems High-capacity RNA-to-cDNA kit (#4387406) per the manufacturer's instructions. cDNA was generated from 250 to 500 ng of RNA using Applied Biosystems High-capacity RNA-to-cDNA kit (#4387406) per the manufacturer's instructions. Quantitative PCR was performed on a QuantStudio 6 Real Time Polymerase Chain Reaction (PCR) system (with Thermo Fisher Design & Analysis Software version 2.6.0) using PowerUp SYBR Green master mix (Applied Biosystems A25776). The PCR phase consisted of 40 cycles of 15 s at 95°C and 1 min at 60°C. Forward and reverse primer sequences for *Tardbp*, *Sort1-WT*, *Sort1-total*, *Sort1-ex17b*, *TARDBP*, *SORT1-WT*, *SORT1-ex17b*, *SORT1-total*, *UNC13A*, *UNC13A* cryptic exon, *STMN2*, truncated *STMN2*, and the reference genes *β-Actin*, *Pgk1*, and *RPLP0* are listed in *Supplementary file 3*. Relative quantification of transcripts was performed using the Pflaff method (*Pfaffl, 2001*) with both *β-Actin* and *Pgk1* as reference genes for mouse tissue and mouse primary neurons, and RPLP0 used as the reference gene for hiPSC-derived neurons. Primers generated for this paper were designed using PrimerBank (*Wang et al., 2012*) and synthesized by Integrated DNA Technologies (IDT).

## Tissue collection, staining, and immunofluorescence

Mice were transcardially perfused with 1× PBS followed by 4% PFA, and brains were removed and post-fixed for 24 hr in 4% PFA. Brains were cryoprotected in 15% sucrose in 1× PBS for 24 hr followed then 30% sucrose in 1× PBS for 48 hr, and then embedded in Fisher Tissue-Plus OCT compound (4585). Forebrain (~Bregma +1.78 mm), midbrain (~Bregma +0.50 mm), and hippocampal (~Bregma −1.94 mm) 10 mm cryosections were collected onto Superfrost Plus charged slides using a Leica CM1950 cryostat.

## Luxol Fast Blue and Cresyl Violet staining

Slides were acclimated to RT and rinsed in distilled water. They were then incubated in 70% ethanol for ~60 hr followed by 95% ethanol for 30 min. Tissues were stained with 0.1% Luxol Fast Blue Solution overnight at 60°C, rinsed in distilled water, and differentiated as needed by 0.05% Lithium Carbonate followed by 70% ethanol. They were then counterstained with 0.1% Cresyl Echt Violet, rinsed in distilled water, and differentiated again by 95% ethanol. Finally, slides were dehydrated in 100% ethanol, cleared in Xylene, and coverslipped with DPX mountant (Millipore-Sigma 44581).

## Immunofluorescence

Slides were acclimated to RT and rinsed in distilled water. Heat-induced epitope retrieval was performed at 120°C using pH 6.0 buffer (Epredia, TA-135-HBL). The tissues were then blocked in 10% NGS for 1 hr. Primary antibodies were applied overnight at 4°C: Mouse Anti-NeuN (clone A60) conjugated to Alexa Fluor 555 (Millipore, MAB377A5), and Rabbit Anti-TDP-43 (Cell Signaling, 89789) or Rabbit anti-Iba1 (Wako, 019-19741) and mouse anti-GFAP conjugated to Alexa Fluor 488 (Cell Signaling, 3655) (*Supplementary file 4*). After rinsing, secondary antibodies were applied for 2 hr at RT: Alexa Fluor 488 Goat Anti-Rabbit IgG (Invitrogen, A32731) or Alexa Fluor 680 Goat anti-Rabbit IgG (Invitrogen 32734). All antibodies were diluted using Da Vinci Green Diluent (Biocare Medical, PD900L). Finally, slides were coverslipped using Fluorogel II with DAPI (Electron Microscopy Sciences, 17985-50).

## Tissue microscopy and image analysis

Brain sections stained with Cresyl Violet (CV) and brain sections labeled with Iba1 and GFAP antibodies were imaged on a Nikon Eclipse Ti2 widefield microscope equipped with a Nikon DS-Fi3 CMOS color camera (for CV imaging) and a pco.edge 4.2Q High QE sCMOS camera (PCO, Kelheim, Germany) using a ×20/0.5 NA Plan Fluor objective and NIS-Elements software (Nikon, Minato City, Tokyo, Japan). Immunofluorescently labeled brain sections were imaged on a Leica SP8X Falcon confocal microscope equipped with hybrid GaAsP detectors using a ×40/1.30 NA Plan Apochromatic (oil) objective and Leica Application Suite X Life Sciences software (Leica, Wetzlar, Germany). Cell counting (density of CV- and NeuN-positive cells), as well as TDP-43 fluorescence intensity, TDP-43 localization measurements, GFAP fluorescence intensity and area, and Iba1 fluorescence intensity and area were performed using CellProfiler (*Carpenter et al., 2006*; *Stirling et al., 2021*). Prior to quantifying TDP-43 immunofluorescence intensity and prior to confocal image segmentation, immunofluorescent images were denoised using NIS-Elements Batch Denoise.ai under default conditions. Prior to quantifying Iba1 and GFAP fluorescence characteristics, images of the necortex and hippocampus were cropped to exclude fluorescent signal from the corpus callosum. Within CellProfiler, the Cellpose 2.0 plugin (*Stringer and Pachitariu, 2022*; *Stringer et al., 2021*) was used to perform neuron identification and the subcellular segmentation of nucleus and cytoplasm within neurons. DAPI fluorescence was used to identify nuclei, and NeuN immunoreactivity was used to identify neuronal nuclei and the surrounding soma. Images were pseudocolored and formatted for publication using Fiji ImageJ (*Schindelin et al., 2012*).

## Statistical analysis

Statistical analysis of RNA sequencing data was performed as described above. For data shown in *Figures 1B, C, 2B, C, 4B, C, E, F , and H, I*, we used linear mixed effects (LME) models with random intercept to allow for an animal or a well specific effect. Restricted maximum likelihood (REML) approach was used for parameter estimation. For *Figures 1B, C, and 2B, C*, wells were used as a 'by-subject' random effect. For *Figure 4B, C*, animals were used as a 'by-subject' random effect. For 4E, F and 4H, I, animal ID and image ID were included as nested random effects. Either genotype or treatment group were included as fixed effects in each LME model. Results are presented as fixed effect estimates, standard errors and 95% confidence intervals (see *Supplementary file 2*). All statistical analyses were performed in R version 4.2.1 (*R Development Core Team, 2022*). Complete case analysis was considered, with $p < 0.05$ determining statistical significance. All other data were analyzed in GraphPad Prism Version 9.4.1 for Windows, GraphPad Software, San Diego, CA, USA, https://www.graphpad.com. For datasets with $n > 10$, outliers were identified and removed (if any) using the ROUT method (*Motulsky and Brown, 2006*) at $Q = 1\%$. Details regarding the statistical

test performed, sample sizes, and what the datapoints and error bars represent can be found in the appropriate figure legends. All statistical tests were two sided. Statistical significance was determined as $p < 0.05$. Supplementary statistical information about data presented in main figures and figure supplements can be found in the accompanying source data files.

## Acknowledgements

We would like to thank Dale Cowley and the UNC Animal Models Core Facility for their creation of the TDP-43$^{K145Q}$ mouse line and the initial sequencing. We also thank Natallia Riddick and the UNC Mouse Breeding and Colony Management Core for their assistance in animal husbandry and breeding. Confocal and Nikon Ti2 widefield microscopy was performed at the UNC Neuroscience Microscopy Core (RRID:SCR_019060), supported, in part, by funding from the NIH-NINDS Neuroscience Center Support Grant P30 NS045892 and the NIH-NICHD Intellectual and Developmental Disabilities Research Center Support Grant P50 HD103573, and core director Michele Itano assisted in image analysis protocol design. We are grateful for the UNC Histology Research Core for their assistance in tissue processing and staining. RNA sequencing was performed by the New York Genome Center. The research reported in this publication was supported by the National Institute on Aging (grant F30AG072786), the National Institute on Neurologic Disorders and Stroke (grants R01NS105981, P30NS045892, F31NS122242), the National Institute of General Medical Sciences (grants 1T32GM133364-01A1, 5T32GM008719-19), the Eunice Kennedy Shriver National Institute of Child Health and Human Development (grants U54HD079124; P50HD103573), and the National Center for Advancing Translational Sciences (grant UM1TR004406) of the National Institutes of Health. We also received support from the Department of Defense (grant AL180038) and the Muscular Dystrophy Association (grant MDA573414).

## Additional information

### Funding

| Funder | Grant reference number | Author |
|---|---|---|
| National Institute of Neurological Disorders and Stroke | R01NS105981 | Todd J Cohen |
| National Institute of Neurological Disorders and Stroke | F31NS122242 | Baggio A Evangelista |
| Eunice Kennedy Shriver National Institute of Child Health and Human Development | U54HD079124 | Jeremy M Simon |
| National Institute of Neurological Disorders and Stroke | P30NS045892 | Jeremy M Simon |
| National Institute on Aging | F30AG072786 | Julie C Necarsulmer |
| Eunice Kennedy Shriver National Institute of Child Health and Human Development | P50HD103573 | Sheryl S Moy |
| National Institute of General Medical Sciences | 1T32GM133364-01A1 | Julie C Necarsulmer |
| National Institute of General Medical Sciences | 5T32GM008719-19 | Julie C Necarsulmer |
| Muscular Dystrophy Association | MDA573414 | Todd J Cohen |

| Funder | Grant reference number | Author |
| --- | --- | --- |
| U.S. Department of Defense | AL180038 | Todd J Cohen |
| National Center for Advancing Translational Sciences | UM1TR004406 | Huijun Jiang Feng-Chang Lin |

The funders had no role in study design, data collection, and interpretation, or the decision to submit the work for publication.

## Author contributions

Julie C Necarsulmer, Conceptualization, Data curation, Formal analysis, Funding acquisition, Validation, Investigation, Visualization, Methodology, Writing – original draft, Writing – review and editing; Jeremy M Simon, Formal analysis, Supervision, Visualization, Methodology; Baggio A Evangelista, Data curation, Methodology; Youjun Chen, Ping Wang, Data curation, Investigation, Methodology; Xu Tian, Investigation, Methodology; Sara Nafees, Validation, Investigation; Ariana B Marquez, Data curation, Investigation, Visualization, Methodology, Writing – review and editing; Huijun Jiang, Data curation, Formal analysis, Investigation, Methodology; Deepa Ajit, Viktoriya D Nikolova, Data curation, Formal analysis, Investigation, Visualization, Methodology; Kathryn M Harper, Data curation, Investigation, Visualization, Methodology; J Ashley Ezzell, Data curation, Supervision, Investigation, Visualization, Methodology, Writing – review and editing; Feng-Chang Lin, Sheryl S Moy, Data curation, Formal analysis, Supervision, Investigation, Visualization, Methodology, Writing – review and editing; Adriana S Beltran, Conceptualization, Data curation, Supervision, Funding acquisition, Visualization, Methodology, Writing – original draft, Writing – review and editing; Todd J Cohen, Conceptualization, Data curation, Formal analysis, Supervision, Funding acquisition, Methodology, Writing – original draft, Writing – review and editing

## Author ORCIDs

Julie C Necarsulmer ⓘ http://orcid.org/0000-0002-0669-2023
Jeremy M Simon ⓘ http://orcid.org/0000-0003-3906-1663
Kathryn M Harper ⓘ http://orcid.org/0000-0002-6297-0032
J Ashley Ezzell ⓘ https://orcid.org/0000-0002-5142-4685
Todd J Cohen ⓘ https://orcid.org/0000-0002-4099-0278

## Ethics

Mice were housed in ventilated microbarrier cages on racks providing HEPA filtered air supply to each cage. Animals were kept on a 12-hr light–dark cycle with access to food and water ad libitum. All animal husbandry, experiments, and procedures were performed in strict compliance with animal protocols approved by the Institutional Animal Care and Use Committee (IACUC) of the University of North Carolina at Chapel Hill (Protocol #21.257).

Reviewer #1 (Public Review): https://doi.org/10.7554/eLife.85921.3.sa1
Reviewer #2 (Public Review): https://doi.org/10.7554/eLife.85921.3.sa2
Reviewer #3 (Public Review): https://doi.org/10.7554/eLife.85921.3.sa3
Author Response https://doi.org/10.7554/eLife.85921.3.sa4

# Additional files

## Supplementary files

• Supplementary file 1. Detailed RNA sequencing results and analysis. (a) Differentially expressed genes (DEGs) in the neocortex of 18-month TDP-43$^{KQ/KQ}$ mice compared to WT. (b) DEGs in the hippocampus of 18-month TDP-43$^{KQ/KQ}$ mice compared to WT. (c) Gene Ontology terms enriched in each of the six groups of DEGs in TDP-43$^{KQ/KQ}$ animals compared to TDP-43$^{wt}$ at 18 months old. (d) Intersections between DEGs in TDP-43$^{KQ/KQ}$ micec compared to DEGs in FTLD-TDP cortex and mouse TDP-43 knockdown. (e) List of clusters within differentially spliced genes (analysis by LeafCutter, LeafViz) in the cortex of TDP-43$^{KQ/KQ}$ mice compared to WT at 18 months of age. (f) List of clusters within differentially spliced genes (analysis by LeafCutter, LeafViz) in the hippocampus of

TDP-43<sup>KQ/KQ</sup> mice compared to WT at 18 months of age.

- Supplementary file 2. Complete detailed results of linear mixed effects model statistical analyses.
- Supplementary file 3. Complete list of oligonucleotides used in the present study.
- Supplementary file 4. Complete list of primary and secondary antibodies used in the present study.
- MDAR checklist

## Data availability

The TDP-43K145Q mouse line is now available at the Mutant Mouse Resource and Research Center (MMRRC) at University of North Carolina at Chapel Hill, an NIH-funded strain repository, with the following identifiers (RRID:MMRRC_068119-UNC). Raw and processed RNA-seq data have been deposited to the Gene Expression Omnibus (GEO) under accession GSE216294. All data generated in this work are included in the manuscript and the supporting files. Any plasmids or lentiviruses generated in this study are available upon request.

The following dataset was generated:

| Author(s) | Year | Dataset title | Dataset URL | Database and Identifier |
|---|---|---|---|---|
| Simon JM, Necarsulmer JC, Cohen TJ | 2022 | A TDP-43 acetylation-mimic mutation that disrupts RNA-binding drives FTLD-like neurodegeneration in a mouse model of sporadic TDP-43 proteinopathy | https://www.ncbi.nlm.nih.gov/geo/query/acc.cgi?acc=GSE216294 | NCBI Gene Expression Omnibus, GSE216294 |

The following previously published datasets were used:

| Author(s) | Year | Dataset title | Dataset URL | Database and Identifier |
|---|---|---|---|---|
| Gerrits E, Giannini L, Brouwer N, Melhem S, Seilhean D, Ber Le, Kamermans A, Kooij G, de Vries H, Boddeke E, Seelaar H, van Swieten J, Eggen B | 2022 | Neurovascular dysfunction in GRN-associated frontotemporal dementia identified by single-nucleus RNA-sequencing of human cerebral cortex | https://www.ncbi.nlm.nih.gov/geo/query/acc.cgi?acc=GSE163122 | NCBI Gene Expression Omnibus, GSE163122 |
| Cleveland DW, Yeo GW | 2011 | Disrupted processing of long pre-mRNAs and widespread RNA missplicing are components of neuronal vulnerability from loss of nuclear TDP-43 (RNA-seq) | https://www.ncbi.nlm.nih.gov/geo/query/acc.cgi?acc=GSE27218 | NCBI Gene Expression Omnibus, GSE27218 |

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
