## [Editor Report · eLife assessment]

Necarsulmer et al describe an interesting new mouse model of TDP-43 proteinopathy in which gene editing was used to introduce a K145Q acetylation-mimic mutation previously shown to impair RNA-binding capacity and induce downstream misregulation of target genes. Mice homozygous for this mutation are **convincingly** shown to display cognitive/behavioral impairment, TDP-43 phosphorylation and insolubility, and changes in gene expression and splicing. This novel mouse model replicates some **important** hallmarks of human frontotemporal lobar degeneration and will be an **important** contribution to the field.

---

## [Referee Report · Reviewer #1 (Public Review)]

With this work, the authors address a central question regarding the potential consequences of post-translational modifications for the pathogenesis of neurodegenerative diseases. Phosphorylation and mislocalization of the RNA binding protein TDP43 are characteristic of ~50% of frontotemporal lobar degeneration (FTLD), as well as >95% of amyotrophic lateral sclerosis (ALS). To determine if acetylation is a primary, disease-driving event, they generated a TDP-43 mutant harboring an acetylation-mimicking mutation (K145Q). Animals carrying the acetylation-mimic mutation (K145Q) displayed key pathological features of disease, including more cytoplasmic TDP43 and impaired TDP43 splicing activity, together with behavioral phenotypes reminiscent of FTLD.

This is a well-written and well-illustrated manuscript, with clear and convincing findings. The observations are significant and emphasize the importance of post-translational modifications to TDP-43 function and disease phenotypes. In addition, the TDP43(K145Q) mice may prove to be a valuable model for studying TDP-43-related mechanisms of neurodegeneration and therapeutic strategies.

Comments on the latest version:

The authors have addressed most concerns. The additional analysis demonstrating a lack of neuron loss is quite different from the original study -- it is good that the authors pursued this question. In addition, new data focusing on native TDP-43 splice targets, rather than the splicing reporter, are excellent.

---

## [Referee Report · Reviewer #2 (Public Review)]

This paper extends prior work demonstrating the importance of K145 acetylation of TDP-43 as a post-translational modification that impacts its RNA-binding capacity and may contribute to pathology in FTLD-ALS. The main strengths of this paper are the generation of a novel mouse model, using CRISPR gene editing, in which an acetylation-mimetic mutation (K to Q) is introduced at position 145. Behavioral, biochemical, and genetic analyses indicate that these mice display phenotypes relevant to TDP-43-associated disease and will be a valuable contribution to the field.

---

## [Referee Report · Reviewer #3 (Public Review)]

Numerous experimental models are phenotyped in this manuscript including mouse neurons, iPSC-derived human neurons, knock-in mice, and knock-in iPSCs. Expression of acetylation-mimic or acetylation-null TDP-43 protein is achieved either with overexpression or CRISPR-Cas9-based knock-in. A complex phenotype is observed including loss of TDP-43 function (reduced autoregulation, increased cryptic splicing) and a gain of TDP-43 (increased insoluble TDP-43 protein). These correlate with downstream neurobehavioral changes which are most consistent with a cortical/hippocampal phenotype without a motor phenotype. Post-translational modifications of disease-associated proteins are thought to contribute to neurodegenerative disease pathogenesis, and this study succeeds in demonstrating that TDP-43 acetylation results in downstream molecular and behavioral phenotypes.

TDP-43 acetylation is a post-translational modification that is known to be associated with TDP-43 inclusions that are characteristic of human diseases. An important strength is the rigorous use of multiple different experimental models (rodent cells, iPSC-derived neurons, mice, overexpression, knock-in) with overall consistent results. Moreover, multiple orthogonal endpoints are presented including histology/cytology/immunostaining, biochemistry, molecular biology, and neurobehavioral assays. As TDP-43 acetylation is known to block RNA binding, these novel cellular and mouse models represent interesting albeit complex tools to study the functional consequences of a partial loss of function. As TDP-43 regulates its own expression (i.e. autoregulation), the complexity lies at least in part due to the loss of RNA binding leading to a functional loss of TDP-43 function which includes the increased expression of the TARDBP transcript and TDP-43 protein.

Conceptually, there is a disconnect in that the mouse model exhibits primarily a cortical/hippocampal phenotype more akin to frontotemporal lobar degeneration with TDP-43 inclusions (FTLD-TDP), while TDP-43 acetylation is only seen in ALS tissues and not in FTLD-TDP tissues because most of the pathologic protein in the latter is N-terminally truncated (i.e. the acetylation site is not present). That being said, there is no mouse model which completely and faithfully recapitulates the human disease, and this mouse model avoids overt overexpression (increased TDP-43 protein expression stemming from altered autoregulation) and avoids the use of synthetic/artificial mutation (such as mutation of the TDP-43 nuclear localization signal).

This revision addresses most of my prior comments including documenting the lack of neurodegeneration in this model, the use of more appropriate statistical methods, and the use of more robust/quantitative aberrant splicing measures. The one thing which would still be helpful is sequencing the top predicted off target genomic loci for their various CRISPR'd models irrespective of whether these loci are exonic or noncoding. Having actual sequencing verification of the lack of mutations at these loci is preferable over relying only on computational likelihood estimates.

---

## [Author Response]

The following is the authors’ response to the original reviews.

**Reviewer #1**
1.1 Fig. 1: A good control for these studies would be a TDP-43 variant with an RRM1 mutation that impairs RNA binding, but not an acetylation mimic (i.e. mutations affecting W113, R151, F147, or F149)

In our original paper (Cohen et al, Nat Commun. 2015 Jan 5; 6:5845), we already characterized TDP-43 acetylation and employed a complete RNA-deficient mutant (F147/149L), as the reviewer suggested. In that original study, this mutant showed maximal RNA binding-deficiency, and therefore we proposed that acetylation mimic mutations represent a comparable RNA-binding deficient variant.

1.2 Fig. 1: time and expression level can influence nuclear TDP-43 puncta formation. It is important that the authors take these into account when measuring puncta number/frequency.

All expression levels and transfection/transduction times were identical across samples. We chose the optimal times to express TDP-43 constructs yet minimize toxicity and found that neuronal transduction at DIV10 and arsenite exposure on DIV14 in mature neurons was optimal.

1.3 Fig. 2: to accurately refer to the nuclear foci as anisosomes, the authors will need to conduct higher-resolution imaging.

We agree with the reviewer and since anisosomes are not well characterized in terms of their relationship to TDP-43 nuclear foci (and may represent only a subset of foci), we have now omitted any mention of anisosomes but instead refer to them in the discussion, where we suggest that TDP-43 K145Q foci may partially represent anisosomes.

1.4 Fig. 2D: it seems as though the splicing reporter should have a fluorescence-based readout (red/green ratio, for instance). Is this the case, and is the ratio informative?

We have now removed the splicing reporter data and replaced this with much more robust data showing RT-qPCR of downstream TDP-43 targets including Sortilin-1 (see the new revised Figure 2E and 3B-I).

1.5 Line 145: "Overall, these results indicate that a single endogenously expressed acetylation-mimic TDP-43(K145Q) mutation is sufficient to alter TDP-43 localization, induce TDP-43 phase separation, and impair splicing in a murine primary neuron culture model." The authors did not assess phase separation in this study. Moreover, it would be more convincing to assess native splice targets of TDP-43 in K145Q primary neurons, rather than an exogenous splicing reporter.

See comment 1.1 above. We have now avoided mentioning phase separation in the main text but mention this as a potential mechanism in the discussion. In addition, we have now evaluated native TDP-43 splice targeted in primary neurons.

1.6 Fig. 4A: is the loss of neurons selective for a specific layer or region of the cortex?

Since we did not observe any gliosis, we have gone back and completely re-evaluated neuronal loss since the concept of neurodegeneration is a critical question in the TDP-43KQ/KQ mice. We do not find any significant neuronal loss in the homozygous TDP-43KQ/KQ mice (see Figure 5).

1.7 Fig. 6: The authors suggest that the large majority of splicing changes are direct results of the TDP43(K145Q) mutation and impaired RNA binding by TDP-43. However, without a direct assessment of TDP43(K145Q) target RNAs in comparison to those of TDP-43(WT), this is only an assumption. Moreover, given the fact that RNA-seq was performed in aged animals, the potential for indirect gene expression changes is very high.

In our original study (Cohen et al, Nat Commun. 2015 Jan 5;6:5845), we showed that the K145Q is severely deficient in RNA binding. In this study, we now show strong evidence that many known targets of direct TDP-43 binding are dysregulated, supporting the expected loss of function if TDP-43 K145Q mutation abrogated RNA binding. Although we have not performed direct RNA binding studies to the Sort1 transcript, for example, other studies have clearly indicated that wild-type TDP-43 binds these targets. We infer that loss of function mutations (i.e., K145Q) impact direct targets of TDP-43. Future studies employing RNA-immunoprecipitation followed by RNA sequencing (RIP-seq) could be useful in this regard and will be required to mechanistically address this point.

1.8 Sup Fig. 8 is very interesting and suggests that any TDP-43 variant that is unable to bind RNA may lead to upregulation of TDP43 RNA and phenotypes similar to those observed n K145Q animals. This is alluded to in the discussion but never specifically tested.

Yes, we agree with this reviewer’s comment. Loss of RNA binding, whether due to acetylation (e.g., K145Q) or otherwise is expected to cause autoregulatory up-regulation of the TARDBP transcript and impact other targets, potentially yielding phenotypes similar to the TDP-43KQ/KQ mice. However, new in vivo models would be needed to prove this point. For example, in the future, we will consider this possibility by characterizing recently identified RNA-binding deficient familial TARDBP mutants (e.g., P112H or K181E).

1.9 The authors should also provide some comment or potential explanation for why TDP43(K145Q) animals show no signs of motor neuron disease.

We now show a moderate level of TDP-43 aggregation and hyper-phosphorylation in spinal cord of mutant mice in Figure 6 – Figure Supplement 3. We also speculate in the discussion why we observe aspects of TDP-43 dysfunction in spinal cord without overt motor phenotypes up until 18 months old.

1.10 Line 79: "However, TARDBP mutations that disrupt RNA binding, and thereby may act in a similar manner to TDP-43 acetylation, have been identified in FTLD-TDP patients." Evidence suggests that the D169G mutation does not interfere with RNA binding. See Furukawa et al., 2016.

We thank the reviewer for pointing this out. We have now removed the D169G mutation from the discussion.

1.11 It is unclear why the authors focused solely on homozygous K145Q animals, rather than heterozygous mice.

We focused initially on homozygous mutant mice to provide better statistical power to detect small effect sizes. However, we have now included a thorough analysis of heterozygous mice including molecular analysis of brain tissue and mouse behavior, as shown in Figure 4 – Figure Supplements 1-2 and Figure 6 – Figure Supplements 1-3.

**Reviewer #2**
2.1 A strength of this paper is the generation of a new mouse model to study the effects of K145 acetylation in TDP-43 proteinopathy. While the authors note an absence of a behavioral phenotype on neuromuscular testing in aged animals, it would be appropriate to include some analysis of spinal cord and skeletal muscle in this initial description of their model. At a minimum, I wonder if there is pathology in the cord (neuron loss, gliosis) or muscle (fiber atrophy) if insoluble p-TDP-43 is detectable in these tissues, and whether dysregulated splicing of TDP-43 target genes (such as shown in Fig 7) occurs at these sites.

See comment 1.9 above. We analyzed TDP-43 aggregation, localization, and splicing in the spinal cord of TDP-43KQ/KQ mice and found mild loss of TDP-43 function that was comparable, though not to same extent, as that seen in hippocampus and cortex. We discussed these findings in the discussion and provide several possibilities for why there are no overt motor phenotypes in these mice. We note that TDP-43 Q331K knock-in mice also have cognitive but no motor deficits, suggesting TDP-43 dysfunction may preferentially (or at least initially) impact cognitive function (White et al, Nat Neurosci. 2018 Apr;21(4):552-563).

2.2 Fig 2: Differences in the splicing reporter are hard to appreciate from the images shown in panel E. Is the quantification shown in panel F corroborated by an analysis of green vs yellow fluorescence or by another method? Quantification of results shown in panel 2G (from 3 biological replicates) should be included.

We have now removed the splicing reporter data in lieu of the more robust RT-qPCR data shown in Figure 2E and 3B-I. We have also now included more biological replicates from our iPSC neuron imaging, as shown in Figure 3A. Due to time and resource constraints, we were not able to quantify the images shown in figure 3A, and reinforce in the text that our statements are qualitative. However, we were able to add quantitative analysis of TDP-43 dysfunction, by detecting genotype-dependent splicing changes in hiPSC neurons, as mentioned above, which strengthens our claim that TDP-43 dysfunction is prominent in this culture modee.

2.3 Fig 4: Differences in NeuN quantification without changes in cresyl violet staining or gliosis are surprising and a bit difficult to understand. Is there confirmation of neuron loss through another metric? Is it possible that NeuN expression is lower in mutants without frank neuron loss? Also, although no significant differences were seen by IF for TDP-43 staining, did IF for phospho TDP-43 show differences? One might expect this to be the case given the biochemical findings in Fig 5.

See comment 1.6 above. After a much more in-depth and rigorous assessment, we find little evidence for neurodegeneration. Given the transcriptome data showing that TDP-43 regulates a subset of synaptic genes, we suggest that synaptic deficits underlie the behavioral phenotype rather than neuronal loss.

Regarding phospho-TDP-43 pathology by immunofluorescence (IF) staining, after much effort, we have not been able to detect phospho-TDP-43 pathology by IF in TDP-43KQ/KQ mice. Currently available phospho-TDP-43 antibodies (including those acquired from collaborators) do not work well to detect endogenous mouse TDP-43 by histology or IF staining, and therefore we are somewhat limited technically. Nonetheless, given the increase in phospho-TDP-43 in the insoluble fractions by western blotting combined with the increase in cytoplasmic TDP-43 via biochemical fractionation, our data suggest that phospho-TDP-43 is the relevant species accumulating in the cytoplasm of TDP-43KQ/KQ mice.

2.4 Fig 5: Probing the NC fractions for phospho TDP-43 would be an interesting addition to support the conclusion that increased cytoplasmic localization of the KQ mutant occurs prior to its phosphorylation.

We agree that this would be an excellent addition to our data. Unfortunately, after rigorous antibody validation experiments, we were not able to find a phospho-TDP-43 antibody that specifically detected phosphorylated TDP-43 and did not cross-react with unphosphorylated TDP-43 in the buffers used for N-C fractionations. We tested phospho-TDP antibodies in RIPA (soluble), Urea (detergent-insoluble), and the N-C fractionation buffers, using samples treated or untreated with lambda phosphatase (to de-phosphorylate TDP-43). Only one antibody reliably detected the phosphorylated TDP-43 and not the lambda phosphatase-treated TDP-43 samples, and only did so in the Urea buffer, which is shown by straight westerns in our manuscript. Because of these technical difficulties with the phospho-TDP-43 antibodies, this was a challenging point to address at the moment. As better phospho-TDP antibodies become available, we hope to be able to address this. We therefore cannot definitively conclude that cytoplasmic phospho-TDP-43 pathology is present in these mice, but nonetheless the total phospho-TDP-43 levels are significantly elevated in urea (insoluble) fractions.

2.5 Fig 1: What quantitative criteria were used to distinguish between puncta and foci, as highlighted in panel A? What is the biological significance of this distinction? From the images in panel A, it is difficult to see the TDP-43 foci in wt and K145R expressing cells.

Although the size of nuclear TDP-43 foci can be quite variable, and we are certainly interested in the biological significance of this parameter, we did not focus this study on size profiles of K145Q-induced foci, only their accelerated formation and abundance. Therefore, in the revised manuscript we chose not to explicitly state any differences in “foci” vs. “puncta” and now refer to all nuclear TDP-43 structures as “foci” (removed the word “puncta” throughout).

2.6 Fig 3: In describing the results of context-dependent fear testing, it is more appropriate to state that significant deficits appeared at 18 months, deleting the word "more" on line 186.

We have deleted the work “more”.

**Reviewer #3**
3.1 Multiple figures (1b, 1c, 2b, 2c, 4b, 4d, 4f, 4g, 4i, 4j) include data with multiple measurements per field of view and multiple fields of view per condition. It appears that each measurement was considered an "n" for ANOVA or t-tests, but the data structure violates the requirement that data points are independent. More rigorous statistical methods such as mixed effect models should be considered (see DOI: 10.1016/j.neuron.2021.10.030) which in many cases provide more statistical power. Mixed effects models are the more appropriate statistical method for much of their data. Should the authors want to reanalyze their data with this method, they can reach out to me for an introduction to this statistical model.

We have now re-evaluated the figures mentioned using linear mixed effects models, similar to what the reviewer has mentioned. The new statistical measurements have been incorporated into the revised Figures 1, 2, and 5 (formerly Figure 4). A description of the statistical methods used is now provided in the revised methods section.

3.2 In the introduction, the authors write "we avoid both TDP-43 overexpression and disruption of autoregulatory genomic elements of the endogenous Tardbp transcript" but they show that autoregulation is altered. So shouldn't the acetylation sites be considered a genomic element that regulates autoregulation?

We agree and have now stated that our knock-in approach avoids disrupting surrounding genomic elements (as could occur with transgenic or gene replacement strategies, for example) in order to retain the native Tardbp gene in its unaltered form.

3.3 Suggest editing the language regarding potential neurodegeneration/neuron loss as the same results could be obtained with tissue volume and/or developmental effects independent of progressive neurodegeneration.

See comments 1.6 and 2.3 above. The language has been edited to reflect no apparent neurodegeneration.

3.4 Sequencing the top predicted off-target loci in CRISPR'd mice and iPSC cell lines would help show the absence of off-target mutations.

We described in the methods how potential off-target effects were avoided. We assessed the likelihood of off-target mutations using prediction algorithms to ensure low likelihood. All of the predicted exonic off-target sites have 4 mismatches, making them extremely unlikely to be mutated.

3.5 The authors describe a subtle shift in electrophoretic mobility of the SORT1 protein band in figure 7d/e, but it is unclear why the entire SORT1 band should be shifted up in mutant mice given that the RNA analysis suggests that WT species (not the cryptically spliced +ex17b) is still the major RNA that is expressed. In addition, others have shown that the WT versus +ex17b bands can be resolved (see DOI: 10.1073/pnas.1211577110). Perhaps knockout/knockdown cells can facilitate by providing a positive control for sizing/separation of Sort1 by immunoblotting.

Please refer to our RNA-seq data shown in Figure 8A. In WT mice, nearly 80% of Sort1 transcripts lack exon17b, while this number drops to 23% in the TDP-43KQ/KQ mice. Therefore, the abnormally spliced +ex17b becomes the dominant transcript in TDP-43KQ/KQ mice. Given the prominent +ex17b inclusion that we are observing at the transcript level, it is not surprising that we mostly observe the up-shifted ex17b-containing Sort1 protein band. We have been unable to resolve two distinct bands by immunoblotting in mouse tissues using multiple Sort1 antibodies, including those used in Prudencio et al Proc Natl Acad Sci U S A. 2012 Dec 26;109(52):21510-5. Nonetheless, the up-shifted Sort1 protein is clearly the abnormal variant, as it becomes destabilized in our mice. Another possibility is that partial loss of TDP-43 function, as we suspect occurs in the TDP-43KQ/KQ mice, may magnify (or enhance) the effects on Sort1 such that the dominant Sort1 variant observed is the +ex17b containing variant. We suspect this to be true since this phenomenon was also observed in the Prudencio et al study (see Figures 1-2 in that study).

3.6 The authors may try to corroborate their CFTR splicing results by examining fluorescence as it appears that the construct allows for analysis of splicing differences using GFP vs mCherry expression. This is a minor point as RNA-seq analysis demonstrates abundant splicing changes in acetylation-mimic expression models.

We have now removed the CFTR splicing data entirely and replaced it with more robust readouts of endogenous TDP-43 splicing targets both in vitro (Figure 2E, 3B-I) and in vivo (Figure 8B-C).

3.7 Should the bars in figure 3d for 1 and 2 min be colored in grey/pink? It is unclear why they are clear and only outlined in color.

This point is clarified in the revised Figure 4D legend. In our cue-dependent conditioned fear testing, the filled bars beyond 2 min represents the presence of the auditory cue (tone) and the period of statistical analysis.

3.8 The statistical test used (Fisher's exact test?) for determining overlap between transcriptome datasets should be stated.

We clarified our comment in the results section to reflect the use of over-enrichment analysis. In the methods section, it reads “Previously published differentially expressed genes from Hasan et al95 and Polymenidou et al96 were retrieved from the respective publications; significant over-enrichments as well as human gene symbol mappings to mouse orthologs were performed using gprofiler2 (g:Orth).”